# ReJump: A Tree-Jump Representation
# for Analyzing and Improving LLM Reasoning

Yuchen Zeng [* 1 2]  Shuibai Zhang [* 1]  Wonjun Kang [* 3 4]  Shutong Wu [1]  Lynnix Zou [1]  Ying Fan [1 2]  Heeju Kim [3]
Ziqian Lin [1]  Jungtaek Kim [1]  Hyung Il Koo [3]  Dimitris Papailiopoulos [1 2]  Kangwook Lee [1 5]

## Abstract

Large Reasoning Models (LRMs) are Large Language Models (LLMs) explicitly trained to generate long-form Chain-of-Thoughts (CoTs), achieving impressive success on challenging tasks like math and programming. However, their underlying reasoning "algorithms" remain poorly understood. To investigate this, we propose *ReJump*, which represents a reasoning trace as a visitation order over nodes in a tree of intermediate problem-solving steps. Transitions between nodes, which we term *jumps*, include adjacent moves that capture behaviors such as calculation, and non-adjacent moves that capture behaviors such as backtracking and verification. ReJump enables analyzing LLM reasoning with diverse metrics that quantify exploration, exploitation, overthinking, forgetting, and verification. Using our proposed LLM agent to extract reasoning traces into ReJump format, we evaluate state-of-the-art LRMs on two tasks and find that models with similar accuracy can exhibit distinct reasoning behaviors, while different tasks favor different reasoning styles (e.g., varying balance between exploration and exploitation). To further understand how learning strategies shape reasoning, we use ReJump to compare distilled LRMs with their teachers, compare CoT-prompted LLMs with LRMs, and examine how reinforcement learning affects reasoning behavior. Finally, we show that ReJump can improve reasoning quality at test time

through strategies such as ReJump-guided Best-of-N selection and prompt selection. Our code is available at `https://github.com/UW-M adison-Lee-Lab/ReJump`.

## 1. Introduction

Chain-of-Thought (CoT) prompting improves the performance of Large Language Models (LLMs) on complex tasks, such as mathematical reasoning. This was achieved either by providing exemplars of step-by-step reasoning (Wei et al., 2022a) or by simply adding "Let's think step by step" to the prompt (Kojima et al., 2022), which encourages the model to decompose problems into intermediate steps, yielding more accurate and interpretable outputs. Recent work goes further by internalizing multi-step reasoning through supervised fine-tuning (SFT) or reinforcement learning (RL), leading to the recent flourishing of Large Reasoning Models (LRMs), LLMs explicitly trained to generate long-form CoT, such as DeepSeek-R1 (Guo et al., 2025), o1 (Jaech et al., 2024), and QwQ-32B (Qwen Team, 2025).

Comparisons among LRMs have so far focused primarily on final-answer accuracy. Yet, models arriving at the same answer may follow very different reasoning strategies (Fig. 1). Recent work has begun to explore other crucial dimensions of reasoning, such as overthinking (Chen et al., 2025) and underthinking (Wang et al., 2025), a comprehensive understanding of reasoning behavior is still lacking. For instance, analyzing how a model balances exploration and exploitation or how much it forgets during reasoning could offer deeper insights into its core capabilities. This motivates the need for tools to systematically analyze and compare reasoning processes, raising the following question:

*How can we represent an LLM's reasoning trace to facilitate a comprehensive analysis and comparison of its internal behaviors?*

A natural way to address this question is through a tree-based representation, which captures the overall structure of reasoning, including planning and action transitions. The value of such frameworks was noted by Wu et al. (2025), who introduced an abstract reasoning tree as a theoretical formalism capable of identifying when a model's thought

---

[*]Equal contribution  [1]University of Wisconsin–Madison, Madison, WI, USA  [2]Microsoft Research, Redmond, WA, USA  [3]FuriosaAI, Seoul, Republic of Korea  [4]Seoul National University, Seoul, Republic of Korea  [5]KRAFTON, Seoul, Republic of Korea. Correspondence to: Yuchen Zeng <yuchen.zeng.1998@gmail.com>, Shuibai Zhang <shuibai@cs.wisc.edu>, Wonjun Kang <kangwj1995@furiosa.ai>.

*Proceedings of the 43$^{rd}$ International Conference on Machine Learning*, Seoul, South Korea. PMLR 306, 2026. Copyright 2026 by the author(s).

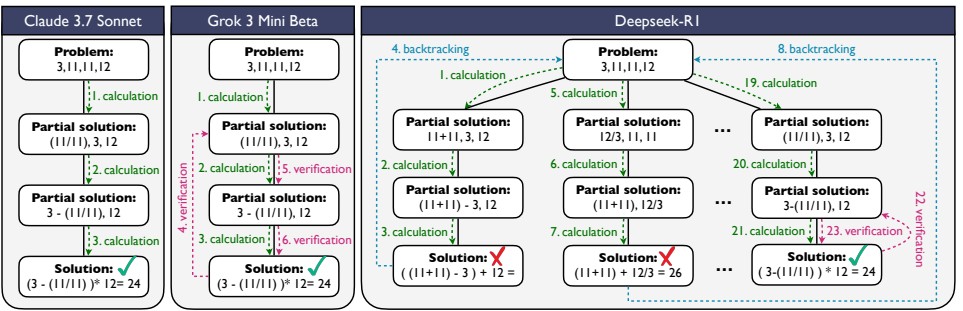

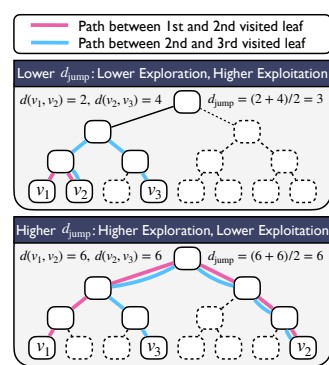

*Figure 1.* **ReJump representations of reasoning traces generated by Claude 3.7 Sonnet, Grok 3 Mini Beta, and DeepSeek-R1 on a Game of 24 problem.** All three models arrive at the same final answer, but their reasoning behaviors differ. Both Claude 3.7 Sonnet and Grok 3 Mini Beta follow a single linear reasoning path; however, Claude 3.7 Sonnet adopts the answer without verification, while Grok 3 Mini Beta verifies it. In contrast, DeepSeek-R1 explores multiple solution paths, exhibiting backtracking and verification.

*Figure 2.* Illustration of how $d_{\text{jump}}$ quantifies the exploration-exploitation trade-off in model reasoning. Shorter jumps indicate local exploration, while larger jumps reflect more global exploration.

process plateaus. However, that work does not operate directly on a reasoning tree and instead relies on hidden states to predict whether a model's thought process plateaus. To bridge this gap, we propose *ReJump*, a tree-jump representation of LLM reasoning that enables comprehensive evaluation and comparison of reasoning traces and can further be leveraged to improve reasoning accuracy. We summarize our main contributions as follows.

**We introduce *ReJump*, a tree-jump representation of LLM reasoning.** ReJump encodes a reasoning trace as a sequence of visited nodes in a hierarchical tree of intermediate problem-solving steps, where the visitation order reflects execution flow and behaviors such as verification, backtracking, and calculation. Unlike a standard tree walk (West, 2001), in which each pair of consecutive nodes is connected by an edge, reasoning traces may include transitions between non-adjacent nodes due to behaviors like backtracking or verification. We therefore term this movement between nodes as a tree *jump*. As shown in Fig. 1, this representation makes it easy to distinguish between different reasoning behaviors. Building on this representation, we define six metrics to quantify reasoning behaviors, along with tree and jump similarity metrics for comparison.

**We design an LLM agent that extracts reasoning traces into the ReJump format.** Given a reasoning trace, our agent, termed *ReJump-Extractor*, performs this in two steps: (i) it first parses the trace into the *tree layer*, where each node represents an intermediate step and each edge encodes the logical dependency between steps; (ii) based on the tree layer, it constructs the *jump layer*, which captures transitions between nodes along with their corresponding action types (verification, calculation, or backtracking). In automatic evaluation, the ReJump representations produced by ReJump-Extractor reach over 0.9 tree and jump similarity with human annotations on Game of 24. In human evaluation, where annotators assess whether each generated

ReJump is correct, ReJump-Extractor achieves over 80% accuracy on MATH-500.

**We utilize ReJump to evaluate and compare reasoning traces across models, tasks, and settings.** We show that models with similar final accuracy can reason in completely different ways, and different tasks also favor different types of reasoning strategies (e.g., varying exploration-exploitation balances). Our analysis further compares reasoning traces across (i) CoT-prompted LLMs and LRMs, showing that LRMs exhibit more deliberate reasoning behaviors such as exploration and verification, and improve performance by generating more diverse solutions, though not necessarily with higher per-attempt accuracy; (ii) distilled models and their teacher LRMs, showing that distilled models inherit reasoning behaviors from their teachers; (iii) varying numbers of in-context reasoning examples, showing that examples more strongly affect reasoning actions than problem decomposition; and (iv) different checkpoints during RL with verifiable reward (RLVR) (Guo et al., 2025), showing that RL reinforces task-preferred reasoning behaviors (e.g., models trained on tasks requiring more exploration exhibit increased exploratory reasoning throughout RL training).

**We leverage ReJump to improve the reasoning of LLMs.** Beyond analyzing the reasoning processes of LLMs, we show that ReJump can enhance performance. ReJump enables Best-of-N (BoN) selection and prompt selection based on desired reasoning properties (e.g., more exploration when helpful). When applied to the Game of 24 benchmark, both methods yield improvements to the pass@1 score, with performance gains ranging from +6.8% to +9.1%.

## 2. Related Work

**Approaches for Reasoning Analysis.** Multiple recent works have introduced approaches to analyze reasoning traces rather than focusing solely final accuracy (Zhou et al.,

2025; Minegishi et al., 2025; Xiong et al., 2025; Feng et al., 2025; Zhang et al., 2025a). Zhou et al. (2025) introduce Landscape of Thoughts (LoT), a visualization method tailored for multiple-choice tasks. It represents each intermediate reasoning step as a vector by computing its perplexity-based distance to all answer options, and then projects these vectors into two dimensions using $t$-SNE for visualization of reasoning trace. They also propose three evaluation metrics: consistency, uncertainty, and perplexity, to analyze model behavior. However, LoT relies on perplexity-based vectors that lack semantic interpretability of the reasoning process. In contrast, several other studies have explored graph-based representations of reasoning traces hence enable quantitative analysis of reasoning traces (Minegishi et al., 2025; Xiong et al., 2025; Feng et al., 2025). Minegishi et al. (2025) perform quantitative analysis via structural properties of graph such as cycles and diameter, and Xiong et al. (2025) propose metrics to evaluate the reasoning's exploration and idea integration behavior. Feng et al. (2025) also propose a graph-based view of reasoning to identify the failed-step fraction and investigate its effect on reasoning accuracy. Zhang et al. (2025a) propose DAG-Math, modeling reasoning traces as directed acyclic graphs (DAGs) and evaluating each trajectory by whether it reaches the correct sink node and by how many nodes in the trajectory fall outside the ancestor set of that sink node. These graph-based frameworks typically encode a reasoning trace in a single layer, where structural dependencies and behavioral transitions are entangled. For example, a backward edge in a flat graph can indicate verification, backtracking, or forgetting, but the graph alone does not specify which behavior occurred. In contrast, ReJump separates the structural hierarchy of partial solutions (the tree layer) from the model's traversal behavior (the jump layer), making metrics such as jump distance, forgetting, overthinking, and verification directly definable. This two-layer view preserves the tree-structured thinking process emphasized by Tree-of-Thought (Yao et al., 2023), while exposing behavioral dimensions that prior graph-based analyses do not capture.

**Empirical Findings on Reasonings.** Prior empirical studies on reasoning typically fall into three categories: (i) limitations in reasoning behavior (Chen et al., 2025; Fan et al., 2025; Wu et al., 2025; Wang et al., 2025), (ii) impact of training algorithms (Yue et al., 2025; Dang et al., 2025), and (iii) factors for effective reasoning (Li et al., 2025). First, a well-known issue of LRMs are overthinking (Chen et al., 2025; Fan et al., 2025; Wu et al., 2025), where models continue unnecessary reasoning even after reaching a correct solution, and underthinking (Wang et al., 2025), where they abandon promising reasoning paths too early, often reflecting excessive exploration. To address overthinking, Wu et al. (2025) introduce thought calibration to dynamically terminate generation by using probes to detect when the model's reasoning tree stops growing. Second, the choice of training

algorithm significantly influences reasoning behavior. Yue et al. (2025); Dang et al. (2025) observe that although RL-trained models outperform base models at small pass@$k$, they merely bias outputs toward rewarded reasoning paths without acquiring new reasoning capabilities, ultimately narrowing reasoning capacity and being surpassed by base models at large $k$. Third, recent work identifies key structural factors that contribute to effective reasoning (Gandhi et al., 2025; Li et al., 2025). Gandhi et al. (2025) highlight behaviors in the base model such as verification and backtracking play a key role in enabling RL training to further develop reasoning ability and improve performance. Similarly, Li et al. (2025) argue that the logical form of reasoning, rather than the content of individual steps, is key to LRM reasoning quality. The ReJump representation facilitates these analyses by systematically capturing overthinking, exploration-exploitation dynamics, and behavioral differences across different experiment settings.

## 3. ReJump: A Tree-Jump Representation of LLM Reasoning

In this section, we introduce the ReJump representation and metrics for analyzing a single tree-jump and comparing pairs of tree-jumps, and describe how reasoning traces are extracted into the ReJump representation in Sec. 4.

### 3.1. Decomposing Reasoning into Tree and Jump Layers

We extract each model-generated reasoning into a ReJump, a two-layer representation that captures both structure and actions of reasoning traces.

- **Tree layer (structure)**: We define a tree $T = (V, E)$, where $V = \{v_i\}_{v=0}^{|V|}$ is the set of nodes and $E$ is the set of edges. Following Yao et al. (2023), each node $v \in V$ represents a partial solution, with the root node corresponding to the initial state containing no solution. An edge $e \in E$ indicates that the parent's partial solution is a direct prerequisite for the child's.

- **Jump layer (action)**: Let $\boldsymbol{i} = (i_0, i_1, \ldots, i_K)$ denote the sequence of reasoning steps, where $i_k$ refers to the index of $k$-th visited node in the tree. The jump starts at $v_{i_0}$ (the root) and ends at $v_{i_K}$ (the final solution). Each transition between consecutive steps $(i_k, i_{k+1})$ is labeled with an action type $\phi_k \in \{\texttt{calc}, \texttt{verify}, \texttt{backtrack}\}$, where $k = 0, \ldots, K-1$. Here, $\texttt{calc}$ refers to generating an intermediate step via calculation or derivation. Both $\texttt{verify}$ and $\texttt{backtrack}$ involve returning to a previously visited node: $\texttt{verify}$ checks its correctness, while $\texttt{backtrack}$ restarts from it to explore an alternative reasoning path. The sequence of actions in the jump layer is denoted by $\boldsymbol{\phi} = (\phi_0, \ldots, \phi_{K-1})$. A jump layer is defined as the pair $W = (\boldsymbol{i}, \boldsymbol{\phi})$, specifying the sequence of visited nodes and the corresponding transitions taken during reasoning.

## 3.2. Quantifying Reasoning Behavior

We define a *derived solution step* as any step in the jump layer that reaches a leaf node via a `calc` transition, thereby contributing to the solution. Steps that visit a leaf solely for verification are excluded, whereas revisiting an already-seen leaf via `calc` still qualifies as a derived solution step.

**Evaluation Metrics.** This representation enables analysis of LLM reasoning behaviors, including solution diversity, exploration–exploitation trade-off, effectiveness in identifying correct paths, frequency of overthinking, forgetting, and verification. These aspects are quantified using the metrics below, computed across all reasonings and their corresponding ReJumps within a task. Each metric below is defined at the instance level and is directly averaged across instances to obtain the task-level score, except for $r_{\text{forget}}$, which is defined only at the task level and described accordingly.

- **Solution Count** ($\#_{\text{solution}}$): Number of distinct solutions, i.e., the total leaf nodes (including incomplete ones).

- **Jump Distance** ($d_{\text{jump}}$, Fig. 2): Measures the exploration-exploitation balance by computing the tree distance (edge count) between the nodes of each consecutive pair of derived solution steps. For a reasoning instance, $d_{\text{jump}}$ is the average distance across all consecutive pairs of derived solution steps within the jump layer.

- **Success Rate** ($r_{\text{success}}$): Fraction of derived solution steps that yield a correct answer. Measures the efficiency.

- **Verification Rate** ($r_{\text{verify}}$): Fraction of all transitions labeled `verify`. Indicates how deliberate and self-critical the reasoning process is.

- **Overthinking Rate** ($r_{\text{overthinking}}$): Fraction of derived solution steps that occur *after* the first correct derived solution step is found, quantifying unnecessary exploration. Quantifies unnecessary exploration and inefficiency.

- **Forgetting Rate** ($r_{\text{forget}}$): This metric is defined only at the task level. A reasoning trace is flagged as *forgetting* if it revisits the same leaf node via `calc`, indicating that the model has re-entered an already visited path; see Fig. 9. $r_{\text{forget}}$ is the proportion of such reasoning traces within the task, highlighting poor memory or state-tracking.

See Sec. A.1 for formal mathematical definition of these metrics. In Sec. A.2, we analyze redundancy among the proposed metrics and show that no metric can be expressed as a deterministic function of the remaining metrics.

**Comparison Metrics.** To assess the similarity between reasoning process produced by different models, we introduce similarity metrics to compare tree and jump representation, respectively. As with the evaluation metrics, all comparison metrics below are defined at the instance level and averaged across instances to obtain task-level scores.

- **Tree Similarity** ($\text{Sim}_T$): Measures the similarity in problem decomposition structure between two reasoning traces by comparing their corresponding trees. These metrics assess whether models adopt similar reasoning structures, without relying on the exact content of individual steps. This choice is justified by the work of Li et al. (2025), which demonstrates that overall logical structure, rather than the specific content at each node, is the primary factor influencing reasoning quality. Given reasoning trees $T = (V, E)$ and $T' = (V', E')$, we compute their Zhang-Shasha Tree Edit Distance (TED) using ZhangShasha dynamic programming algorithm (Paaßen, 2018), which measures the minimum number of edit operations to transform one tree into another. Note that we do not consider relabeling and only use insertions and deletions, since node semantics are not part of this setting. The tree similarity is then defined as $\text{Sim}_T(T, T') = 1 - \text{TED}(T, T') / \max(|V|, |V'|)$.

- **Jump Similarity** ($\text{Sim}_J$): Measures the similarity in action transition patterns between the jumps derived from two reasoning traces. For each reasoning jump $W = (\boldsymbol{i}, \boldsymbol{\phi})$, we construct a $3 \times 3$ transition probability matrix $P$, where $P_{a,b}$ is the empirical probability of transitioning from action $a$ to $b$, with $a, b \in \{\texttt{calc}, \texttt{verify}, \texttt{backtrack}\}$. Given two jump layers $W, W'$ with transition probability matrices $P$ and $P'$, we define their similarity as $\text{Sim}_J(W, W') = 1 - \text{JS}(P \| P')$, where JS is the Jensen-Shannon divergence, a symmetric and bounded variant of KL divergence defined as $\text{JS}(P \| P') = \frac{1}{2} \text{KL}(P \| \frac{1}{2}(P + P')) + \frac{1}{2} \text{KL}(P' \| \frac{1}{2}(P + P'))$. Higher $\text{Sim}_J$ values indicate greater alignment in action transition behavior.

Note that our current tree and jump similarity metrics capture only the logical structure and action-transition distributions. This design keeps computation simple but can overlook important differences: two models may produce identically structured trees while differing in semantic content, and a perfect jump-similarity score may still mask distinct temporal patterns. For example, one model may perform all derivations before verifying, while another alternates between computation and verification. Incorporating semantic information and temporal dynamics is therefore a promising direction of future work.

# 4. ReJump-Extractor: Extracting CoTs into ReJump Format

We introduce ReJump-Extractor, an LLM agent for extracting reasoning traces into ReJump format via two steps:

**(1. Tree Layer Extraction)** We use Gemini 2.5 Pro (Google Deepmind, 2024) to extract both the tree and the jump representations from each reasoning trace. Given the original problem input and the model-generated reasoning, we prompt LLM to produce a JSON object that encodes the reasoning tree. This JSON is a dictionary where each key corresponds to a node index, and each value contains three

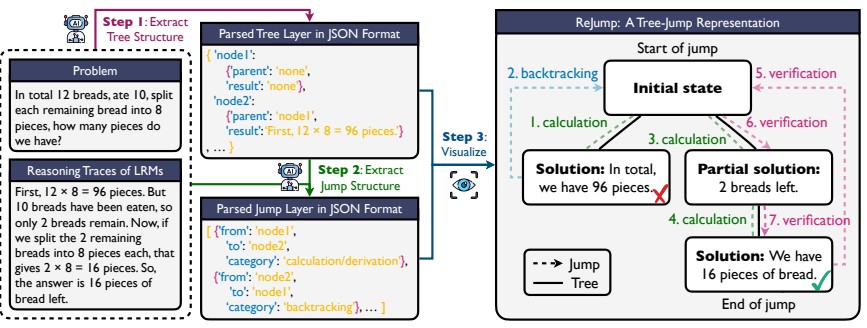

Figure 3. **Illustration of how ReJump-Extractor converts a reasoning trace into the ReJump representation for a math word problem.** This example is crafted for demonstration purposes. Nodes represent partial solutions, and tree edges indicate that the parent nodes serve as prerequisite for child nodes. Dashed arrows show how the reasoning moves between nodes, with each transition corresponding to an action type (`calc`, `verify`, or `backtrack`), and color-coded accordingly.

| | $\text{Sim}_T$ | $\text{Sim}_J$ |
|---|---|---|
| ReJump-Extractor | .943 | .940 |

Table 1. Average alignment between ReJump-Extractor outputs and GT ReJump on Game of 24.

| Reasoning Model | Pass@1 | Pass@2 | Pass@3 |
|---|---|---|---|
| DeepSeek-R1 | 81% | 87% | 90% |
| QwQ-32B | 80% | 83% | 94% |

Table 2. Human evaluation of decomposition of reasoning traces via ReJump accuracy on MATH500.

fields: "problem" (the subproblem addressed at that node), "parent" (the index of the prerequisite node whose partial solution this node builds upon), and "solution" (the result corresponding to the subproblem). For the root node, all three fields are either left empty or labeled as "initial state." **(2. Jump Layer Extraction)** We parse the JSON dictionary to construct the tree structure. Then, using the original input, the full reasoning, and the generated tree JSON as context, we prompt LLM again to extract the jump layer. The jump layer is represented as a JSON list, where each entry describes a transition between nodes, with fields "from," "to," and "category" indicating the source node, target node, and transition type (e.g., calculation, verification, or backtracking). We use this information to visualize the full reasoning trajectory overlaid on the constructed tree.

A visualization of how ReJump-Extractor extracts CoTs into the ReJump format is shown in Fig. 3. The extraction prompt is in Sec. B.1, and additional ReJump examples from real MATH-500 and Game of 24 traces are in Sec. B.2. Because ReJump-Extractor is a diagnostic module, it need not be applied to every model output in deployment. In practice, it can be run on representative traces or sampled checkpoints, and the extractor itself can be replaced by cheaper open-source models or a smaller model fine-tuned for tree-jump extraction.

### 4.1. Assessing the Reliability of ReJump-Extractor

To evaluate the reliability of ReJump-Extractor, we use two complementary methods: (i) automatic evaluation and (ii) human assessment. We apply automatic evaluation to Game of 24 for two reasons. First, it is straightforward to design reasoning traces for Game of 24 with controlled properties such as higher exploration or exploitation, allowing us to curate a dataset of traces paired with ground-truth trees for automated verification. Second, without this dataset, manual verification would be infeasible because Game of 24 traces often involve extensive exploration with many nodes. For MATH500, which is substantially easier for humans

to verify and for which constructing a ground-truth dataset would require significantly more effort, we instead rely on manual inspection of the extracted ReJump representations. **Automatic Evaluation.** *(Dataset)* The dataset contains 82 reasoning traces, each paired with a manually constructed and verified ground-truth (GT) ReJump representation. Constructing the dataset involves two steps. First, we create ReJump examples with varying properties: low and high exploration, low and high verification, with and without forgetting, and with and without overthinking. The resulting trees range from 4 to 20 nodes. We then use Gemini 2.5 Pro to generate natural-language reasoning for each ReJump representation, followed by human review and correction, which produces the final paired reasoning traces and GT ReJump representations. *(Results)* We use ReJump-Extractor to process the reasoning traces and compare its outputs with the ground-truth data, reporting tree and jump similarities in Tab. 1. We find that ReJump representations extracted by ReJump-Extractor using Gemini 2.5 Pro align closely with the ground truth, confirming the reliability of our approach. We further verify in Tabs. 10 and 11 that extractor errors do not meaningfully change downstream metric-based conclusions: all six metrics preserve ground-truth trace rankings with Pearson correlation at least 0.91, and controlled segmentation or action-label perturbations induce modest relative errors. Comparison of Gemini 2.5 Pro with Claude 3.7 Sonnet as the base model for ReJump-Extractor is provided in Sec. B.3. While Claude 3.7 Sonnet also performs well, Gemini 2.5 Pro remains the preferred choice due to its higher accuracy.

**Human Assessment.** We perform human evaluation of the ReJump extracted by ReJump-Extractor from reasoning traces generated by DeepSeek-R1 and QwQ-32B on MATH500. For each prompt, we collect two reasoning traces, one from each model. For every trace, we run ReJump-Extractor three times to obtain three ReJump representations, then manually inspect each for correctness. This allows us to compute pass@1, pass@2, and pass@3 by considering the first representation only, the first two,

and all three. Results are in Tab. 2. A single extraction (pass@1) already yields high reliability, with accuracies of 81% for DeepSeek-R1 and 80% for QwQ-32B, and multiple extractions further improve performance, surpassing 90% accuracy by pass@3.

We show that ReJump-Extractor is robust to prompt variations (Sec. B.4) and compare it to a simpler baseline that directly uses LLM reasoning traces without converting them into ReJump (Sec. B.5). Metric-level checks in Tabs. 10 and 11 further show that the extracted ReJumps preserve ground-truth metric rankings and remain stable under controlled segmentation and action-label perturbations. This explicit representation is also more reliable than asking an LLM to infer metrics directly from raw traces: in Sec. B.5, direct querying is less accurate for metrics that do not require graph structure, and graph-defined metrics such as $d_{jump}$ and $r_{verify}$ cannot be computed faithfully without the extracted tree and jump layers.

## 5. Analyzing LLM Reasoning with ReJump

Building on the ReJump representation introduced in Sec. 3 and ReJump-Extractor introduced in Sec. 4, we analyze the reasoning structures of five state-of-the-art LRMs across two datasets. The two datasets we consider are: (i) MATH-500 (Lightman et al., 2024), a widely used benchmark for mathematical reasoning, and (ii) Game of 24, a task that requires strong planning and enables clear inspection of model behavior in terms of exploration, exploitation, and related strategies, as previously adopted by Yao et al. (2023). Unless otherwise specified, all experiments use a decoding temperature of 0 to ensure deterministic and reproducible outputs. In addition, we examine the effect of reasoning examples and alternative decoding strategies on reasoning behavior in Sec. C.5, and extend the task-level comparison to Sudoku, ZebraLogic, and AIME'26 in Fig. 13.

### 5.1. Comparing Reasoning Structure across State-of-the-Art LRMs and Tasks

This experiment focuses on the state-of-the-art LRMs for which we have access to intermediate reasoning traces: DeepSeek-R1 (Guo et al., 2025), QwQ-32B (Qwen Team, 2025), Grok 3 Mini Beta (xAI, 2025), Phi-4-reasoning-plus (Abdin et al., 2025), and Claude 3.7 Sonnet (Anthropic, 2025). Since Claude 3.7 Sonnet uses a fixed temperature of 1 in thinking mode, and Phi-4-reasoning-plus performs significantly better with temperature 1 than with 0, we set the decoding temperature to 1 for all models in this experiment. The more detailed setup and results for DeepSeek-R1, QwQ-32B, and Grok 3 Mini Beta under temperature 0 are deferred to Sec. C.1. We generate reasoning traces and final answers for both MATH-500 and Game of 24 using each model, and analyze their reasoning behaviors through ReJump. Fig. 4 reports pass@1 and the six reasoning metrics.

| Dataset | $\#_{solution}$ | $d_{jump}$ | $r_{success}$ | $r_{verify}$ | $r_{overthinking}$ | $r_{forget}$ |
|---|---|---|---|---|---|---|
| MATH-500 | .0177 | .0541 | **.8548** | .0173 | .0219 | .0341 |
| Game of 24 | .1402 | **.2742** | **.2146** | .0980 | .1413 | .1317 |

*Table 3.* **Contribution of reasoning metrics to solution correctness on MATH-500 and Game of 24 for DeepSeek-R1, Grok 3 Mini Beta, QwQ-32B, Phi-4-reasoning-plus, and Claude 3.7 Sonnet.** "Contribution" is quantified by the feature-importance scores produced when an XGBoost model is trained on six reasoning-evaluation metrics to classify each solution as correct or incorrect; the model attains accuracies of 0.9197 (MATH-500) and 0.7980 (Game of 24) versus majority-class baselines of 0.6928 and 0.6768. Metrics with importance scores > 0.2 are boldfaced. Game of 24 emphasizes exploration (i.e., $d_{jump}$), while MATH-500 emphasizes exploitation (i.e., $r_{success}$).

**Comparison across Tasks.** MATH-500 and Game of 24 differ in the structure and demands of reasoning. MATH-500 problems are typically deterministic, with only one or two valid solution paths, encouraging focused, step-by-step reasoning. In contrast, Game of 24 requires generating diverse arithmetic expression to reach the target number, promoting trial-and-error and exploratory strategies. This difference is evident in the results: as shown in Fig. 4, all five models yield a much lower $\#_{solution}$ on MATH-500 compared to Game of 24. The $d_{jump}$ is also substantially higher for Game of 24. Meanwhile, MATH-500 shows much higher success and verification rates, likely due to its proof-like structure that favors thorough verification and local exploitation over broad exploration. To systematically analyze which reasoning behaviors contribute most to final accuracy, we compute the feature importance of six reasoning metrics (Tab. 3). We find that $r_{success}$ has the strongest impact on pass@1 for MATH-500, while both $d_{jump}$ and $r_{success}$ are key predictors of pass@1 on Game of 24. This task-level contrast is not limited to these two benchmarks: Fig. 13 extends the comparison to Sudoku, ZebraLogic, and AIME'26, showing that ReJump captures distinct reasoning profiles across proof-like, search-heavy, logic, and competition-level math tasks.

**Comparison across LRMs.** Among the five models, DeepSeek-R1 and Grok 3 Mini Beta achieve the highest pass@1, followed by QwQ-32B, with Phi-4-reasoning-plus and Claude 3.7 Sonnet lagging behind. Although DeepSeek-R1 and Grok 3 Mini Beta reach similar final accuracy, their reasoning behaviors differ substantially. Compared to DeepSeek-R1, Grok 3 Mini Beta adopts a narrower approach: it explores fewer paths and makes shorter jumps, yet reaches correct solutions more efficiently, as reflected in its higher success rate. By contrast, DeepSeek-R1 engages in broader exploration, producing more candidate solutions and making longer jumps, though at the cost of a slightly lower success rate. Despite this, it ultimately achieves accuracy comparable to Grok 3 Mini Beta. QwQ-32B exhibits less exploration than DeepSeek-R1 and a lower success

rate than Grok 3 Mini Beta, resulting in worse performance compared to both. Phi-4-reasoning-plus and Claude 3.7 Sonnet perform the worst among all models, with even lower levels of exploration and success rates. Notably, Claude 3.7 Sonnet exhibits the least deliberate reasoning behavior, as both exploration ($\#_\text{solution}$ and $d_\text{jump}$) and $r_\text{verify}$ are low. Phi-4-reasoning-plus demonstrates slightly more deliberate reasoning behaviors, with higher exploration than Claude 3.7 Sonnet, which contributes to its relatively better performance on Game of 24. All models exhibit overthinking, which is an issue previously observed in LRMs (Chen et al., 2025; Yang et al., 2025), as well as forgetting, both of which reflect inefficient reasoning behaviors.

> *Findings*: *(i) Task characteristics shape and favor distinct reasoning behaviors; (ii) models achieving comparable accuracy may employ distinct reasoning strategies. (iii) overthinking and forgetting are prevalent across LRMs.*

### 5.2. Comparing Reasoning Structure: LLMs vs. LRMs

While LRMs are optimized for multi-step reasoning, general-purpose LLMs can still reason effectively when prompted (e.g., with CoT). Their differing training objectives lead to distinct reasoning behaviors. Using ReJump, we compare the reasoning behaviors of DeepSeek-V3 with CoT and DeepSeek-R1. Fig. 5 shows results on Game of 24. Compared to Yue et al. (2025), which argue that RL favors high-reward paths with less exploration by showing that RL-trained LRMs outperform base models at low $k$ in pass@$k$, but underperform at high $k$, our results show that LRMs outperform LLMs in pass@1, but not by favoring high-reward paths. Instead, their success rates are similar, and gains stem from generating more solutions. In fact, LRMs also exhibit more exploratory reasoning, reflected in higher jump distances and more frequent shifts in approach. Note that this does not contradict Yue et al. (2025), as their analysis considers exploration across samples, while ours focuses on single-trace exploration. Lastly, LRMs show higher verification, overthinking and forgetting.

> *Findings*: *(i) LRMs achieve higher pass@1 by generating more numerous and diverse solutions, despite not necessarily improving per-attempt accuracy. (ii) in contrast, LRMs demonstrate more reasoning behaviors, such as more exploration and verification.*

The deferred results for Qwen2.5-32B with CoT vs. QwQ-32B, together with the MATH-500 results for both model settings in Sec. C.2, further support our findings.

### 5.3. Impact of Distillation on Reasoning Structure: Comparing Teacher and Distilled Models

Model distillation transfers the capabilities of large LRMs to smaller, more efficient models (Guo et al., 2025), often preserving task performance. However, its effect on un-

| Reference Model | Metric | | | |
|---|---|---|---|---|
| | $\text{Sim}_T$ (Teacher, ·) | | $\text{Sim}_J$ (Teacher, ·) | |
| | Base | Distilled | Base | Distilled |
| MATH-500 | .715 | **.728** | .771 | **.878** |
| Game of 24 | .360 | **.426** | .873 | **.905** |

*Table 4.* **Tree similarity ($\text{Sim}_T$) and jump similarity ($\text{Sim}_J$) between each model and the teacher model.** Base: Qwen2.5-14B; Distilled: DeepSeek-R1-Distill-Qwen-14B; Teacher: DeepSeek-R1. Both metrics improve after distillation, showing that distilled model more closely replicates the teacher's reasoning structure.

derlying reasoning structure and actions remains unclear. To investigate this, we compare three model types (base, teacher, and distilled) at two scales: the 14B group uses Qwen-2.5-14B and DeepSeek-R1-Distill-Qwen-14B, while the 32B group uses the 32B counterparts of the same models, with DeepSeek-R1 as the teacher in both cases.

We report the similarity to the teacher model before and after distillation for the 14B comparison group in Tab. 14; results for the 32B group and detailed per-metric comparisons for both groups are deferred to Sec. C.3. The results show that the distilled model consistently moves closer to the teacher in both tree similarity ($\text{Sim}_T$) and jump similarity ($\text{Sim}_J$). Further analysis in Sec. C.3 confirms that distilled models inherit reasoning behaviors from teacher LLMs, including broader exploration, verification, and backtracking, though success rates are not improved. In some cases, the inherited reasoning style can even reduce success rate relative to the base model, especially on tasks where correctness depends more on reliable local derivation than broad exploration.

> *Finding: Distilled models inherit reasoning behaviors from teacher models, as evidenced by gains in both tree and jump similarities.*

In Sec. C.3, we further compare the distilled model (DeepSeek-R1-Distill-Qwen-32B) with the RL-trained model (QwQ-32B) as an initial exploration of how SFT and RL differ in their impact on reasoning behavior, and show that the RL-trained model appears to exhibit more deliberate reasoning behaviors, including increased exploration, verification, and overthinking.

### 5.4. Impact of Reasoning Examples on Reasoning Structure

Reasoning examples are often used to improve final accuracy, but their effect on the structure of reasoning is less clear. We therefore vary the number of in-context examples from DeepSeek-R1 and measure how the resulting traces change in both tree similarity and jump similarity. As shown in Fig. 6, adding more examples does not consistently improve pass@1 or tree similarity, but it steadily increases jump similarity. This suggests that examples mainly teach models to imitate fine-grained reasoning actions, such as verification, calculation, and backtracking, rather than changing high-level problem decomposition. Full results on both MATH-500 and Game of 24 appear in Sec. C.4.

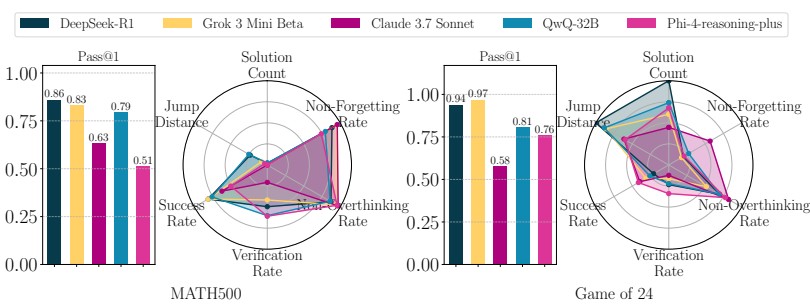

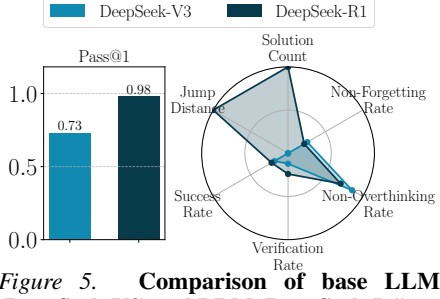

*Figure 4.* **Reasoning performance of state-of-the-art LRMs on MATH-500 and Game of 24.** Bar plots show final accuracy (pass@1), while radar plots summarize six reasoning metrics. Metrics are normalized across models and datasets. Non-forgetting and non-overthinking rates are reported. Models with similar final performance can exhibit substantially different reasoning behaviors.

*Figure 5.* **Comparison of base LLM (DeepSeek-V3) and LRM (DeepSeek-R1) on pass@1 and reasoning metrics for the Game of 24.** Despite similar $r_{success}$, LRM exhibits more exploration, verification, overthinking, and forgetting behaviors.

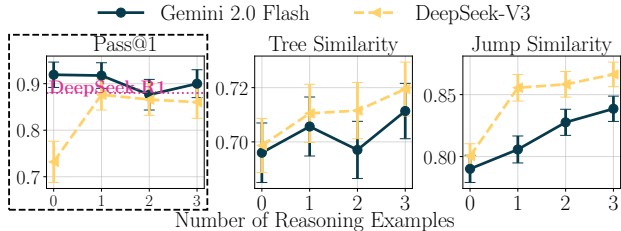

*Figure 6.* **Effect of in-context reasoning examples on MATH-500.** More examples increase jump similarity to DeepSeek-R1, while accuracy and tree similarity do not improve consistently.

> *Finding: In-context reasoning examples affect reasoning actions more consistently than they affect problem decomposition or final accuracy.*

## 5.5. Evolution of Reasoning Dynamics Under RL

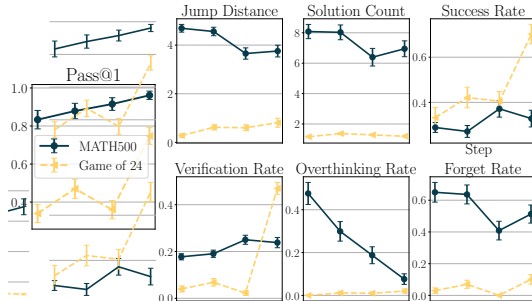

*Figure 7.* **Evolution of reasoning metrics during RL.** We use Qwen3-1.7B for Game of 24 and Qwen3-8B for MATH500, applying DAPO to enhance reasoning performance without SFT, with max sequence length set to 2048. RL encourages more exploratory reasoning on Game of 24, as indicated by increased jump distance, while promoting more exploitative behavior on MATH500, reflected by higher success rates and reduced jump distance. These indicate that RL-induced improvements align with the inherent reasoning characteristics of each task. Note that our pass@1 computation also accounts for the correct formatting.

Guo et al. (2025) demonstrates the effectiveness of RLVR in enhancing the reasoning capabilities of standard LLMs. In this experiment, we use the ReJump representation to visualize the evolution of models' reasoning behavior during RL. We apply DAPO (Yu et al., 2025) using Qwen3-8B on

MATH-500 and Qwen3-1.7B on Game of 24. Each model is evaluated at four checkpoints, corresponding to one-quarter intervals throughout training. This experiment is intended as an analysis use case for ReJump, rather than a comprehensive study of RL scaling. The two model-task pairs are chosen to produce interpretable reasoning dynamics under available compute, and the conclusion concerns behavioral adaptation during RL rather than model-size effects.

As discussed in Sec. 5.1, MATH-500 favors higher success rates, indicating stronger exploitation, whereas Game of 24 benefits from both increased jump distance and higher success rates, reflecting a need for balanced exploration and exploitation. The results in Fig. 7 show that RL progressively shapes the models reasoning dynamics to match these task-specific requirements: promoting more exploitative reasoning (i.e., higher success rates) on MATH500 and both greater exploration (i.e., higher jump distance) and higher success rates on Game of 24.

> *Finding: RL progressively aligns a model's reasoning behavior with the demands of the target task.*

## 6. Enhancing LLM Reasoning with ReJump

We further demonstrate in this section that ReJump can also be leveraged to enhance LLM reasoning.

A key advantage of ReJump is that they enable selecting outputs with desired reasoning characteristics, as measured by the six metrics, without requiring ground-truth solutions. We focus on the Game of 24 because its difficulty depends less on an LLM's raw capability (e.g., success rate) and more on its reasoning behavior (e.g., exploration), which can be more effectively improved by enhancing reasoning patterns. As established in Sec. 5.1, Game of 24 benefits from more explorative reasoning. Therefore, our experiments in this section center on this task instead of MATH500. All reported results are averaged over three random seeds.

### 6.1. Best-of-N Selection with ReJump

Accordingly, we consider two strategies that utilize ReJump for ensembling the result: (i) apply a Best-of-N (BoN) strat-

| Model | Method | pass@1 | $d_{\mathbf{jump}}$ |
|---|---|---|---|
| | MV | 0.76 | 4.20 |
| | Longest Trace | 0.76 | 4.89 |
| QwQ-32B | Most Solutions | 0.74 | 4.34 |
| | Weighted MV w. ReJump | 0.80 | 5.09 |
| | BoN w. ReJump | **0.82** | **5.70** |
| | MV | 0.77 | 3.32 |
| | Longest Trace | 0.77 | 4.71 |
| Phi-4-reasoning-plus | Most Solutions | 0.77 | 4.31 |
| | Weighted MV w. ReJump | 0.81 | 4.02 |
| | BoN w. ReJump | **0.84** | **5.53** |

*Table 5.* **Performance of Majority Vote (MV), simple trace-level heuristics, weighted MV with ReJump, and Best-of-N (BoN) with ReJump on Game of 24 using QwQ-32B and Phi-4-reasoning-plus.** Across both pass@1 and $d_{\mathrm{jump}}$, BoN with ReJump performs the best.

| Model | Prompt | pass@1 | $d_{\mathbf{jump}}$ |
|---|---|---|---|
| QwQ-32B | Default | 0.73 | 4.09 |
| | ReJump-chosen | **0.78** | **4.28** |
| Phi-4-reasoning-plus | Default | 0.76 | 3.42 |
| | ReJump-chosen | **0.82** | **3.98** |

*Table 6.* **Comparison of performance between the default prompt and the prompt selected by ReJump from four candidate prompts on Game of 24.** The ReJump-chosen prompt yields better performance.

egy: generate multiple responses and use ReJump to select the one exhibiting the highest exploration (i.e., the largest jump distance, $d_{\mathrm{jump}}$), (ii) use weighted majority vote (MV), where each response is assigned a weight proportional to the jump distance associated with that response.

We consider QwQ-32B and Phi-4-reasoning-plus in this experiment since their performance is relatively limited and has more room for improvement compared to other LRMs, as shown in Sec. 5.1. We set $N = 3$, which means that for each prompt we generate 3 responses; BoN with ReJump chooses the trace with the highest jump distance, while weighted MV with ReJump assigns weights proportional to each response's jump distance. As baselines, we use MV, selecting the longest trace, and selecting the trace with the most solution attempts. The comparison in Tab. 5 shows that BoN with ReJump outperforms these answer-level and trace-length heuristics. Unlike reward-model selection, this use of ReJump does not require task-specific preference labels or reward-model training. It instead uses a lightweight task-level prior about which reasoning behavior is desirable, such as greater exploration for Game of 24. In Sec. D, we extend our analysis to two additional datasets requiring different reasoning characteristics: for Sudoku and ZebraLogic, selecting lower-$d_{\mathrm{jump}}$ traces improves pass@1. This confirms that ReJump does not prescribe a universal selection rule; the useful reasoning behavior depends on the task.

### 6.2. Prompt Selection with ReJump

Another natural application of ReJump's comprehensive measurement is prompt selection. There are several ways

to leverage ReJump for this purpose. For efficient reasoning, one can design multiple prompts, test them on a development set, and use ReJump to evaluate the generated responses' reasoning behavior. The prompt that achieves a higher success rate and lower overthinking rate can be selected. For search-heavy tasks, such as maze solving or the Game of 24, ReJump can help identify the prompt that provides the best exploration-exploitation tradeoff.

In this experiment, we focus on prompt selection for improving exploration in the Game of 24. We consider QwQ-32B and Phi-4-reasoning-plus. We design four prompts (see Sec. D.2 for details) intended to encourage broader exploration and use ReJump to select the one yielding the highest jump distance. Tab. 6 shows that without ground-truth labels, prompt selection guided by ReJump improves the performance of both models on the Game of 24.

## 7. Discussion

ReJump represents reasoning traces through a tree layer for partial-solution dependencies and a jump layer for traversal behavior. This decomposition enables process-level metrics for exploration, success, verification, overthinking, and forgetting, making it possible to compare models and tasks beyond final-answer accuracy. Across model comparison, distillation, in-context examples, RL, BoN selection, and prompt selection, the same representation exposes both diagnostic differences and actionable improvement signals.

**Limitations and Future Work.** Several limitations remain. *First*, ReJump-Extractor relies on a separate capable LLM to process each reasoning trace, which makes it costly to deploy at scale or during training. This bottleneck can be removed by replacing the extractor with a cheaper open-source model such as Qwen3 or Kimi K2, and even more so by training a small specialized extractor on curated annotations. *Second*, our tree and jump similarities capture only structural and transition-level information, and can therefore mask semantic and temporal differences between traces; for instance, two models may produce identically structured trees yet differ in the content of each step or in the order of execution, such as verifying only at the end rather than interleaving verification throughout. This can be addressed by enriching the similarity metrics with semantic and temporal components, for example by adding an LLM-based content comparison at corresponding nodes and an alignment-based score that penalizes mismatched timing of reasoning actions. *Finally*, applying ReJump to a new task still involves some manual effort, namely choosing the partial-solution granularity and designing task-specific extraction prompts. This can be largely automated using a small labeled development set to pick the most predictive ReJump metric for the task, and ultimately by learning the granularity and prompt design directly from data, which would remove the manual loop entirely.

## Impact Statement

This paper presents a method for representing and analyzing the reasoning processes of large language models, with the goal of advancing understanding of machine learning reasoning behavior. By enabling more systematic analysis of exploration, verification, and inefficiency in model reasoning, this work may contribute to the development of more reliable and interpretable learning systems. The proposed approach is intended as an analytical tool and does not introduce new deployment scenarios or capabilities beyond existing large language models. We do not foresee significant negative societal impacts beyond those already associated with current machine learning technologies.

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

# Appendix

The appendix is organized as supplements to the main text. Sec. A supplements Sec. 3 with the formal ReJump construction and metric definitions. Sec. B supplements Sec. 4 with extraction prompts, real-trace examples, model comparisons, and robustness checks. Sec. C supplements Sec. 5 with additional behavioral comparisons across tasks, models, distillation, examples, and decoding strategies. Sec. D supplements Sec. 6 with additional Best-of-N and prompt-selection details. Secs. E and F report compute resources and LLM usage.

**Contents**

# A. Extended Sec. 3: ReJump

This section expands the representation introduced in Sec. 3. We first make the tree and jump layers precise, then give the formal definitions of each metric used throughout the experiments. To make the discussion self-contained, we also repeat some key content from the main text.

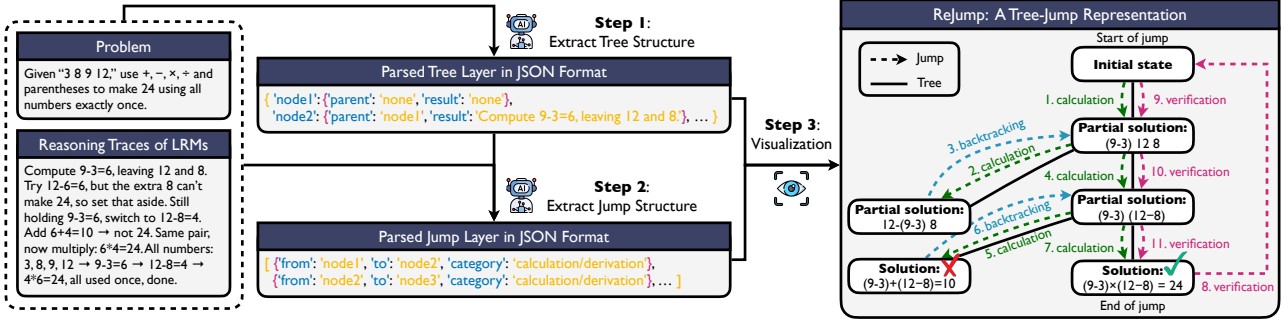

*(a)* ReJump for a Game-of-24 instance. Partial solution here is defined as an intermediate arithmetic expression.

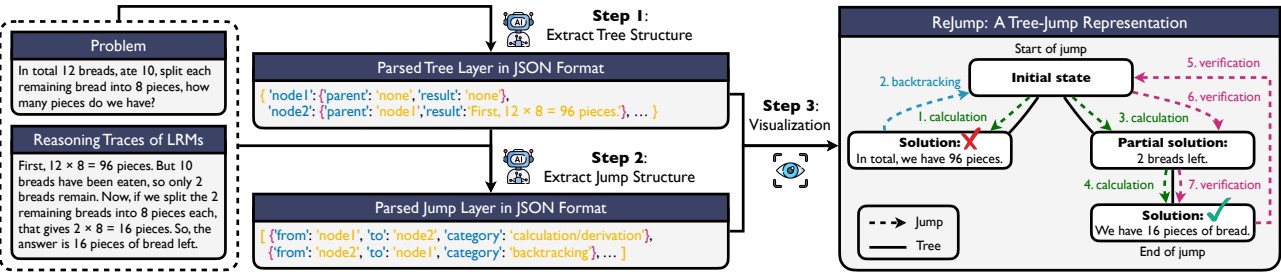

*(b)* ReJump for a math word problem. Partial solution here is defined as an intermediate computed result.

*Figure 8.* **Illustration of ReJump representation of reasoning traces for two different tasks.** These examples are crafted for demonstration purposes. Nodes represent partial solutions, and tree edges indicate prerequisite relationships. The dashed jump traces the solvers reasoning trajectory, with transitions labeled by action type: calculation/derivation, verification, or backtracking, highlighted in different colors.

- **Tree layer (structure)**: We define a tree $T = (V, E)$, where $V = \{v_i\}_{v=0}^{|V|}$ is the set of nodes and $E$ the set of edges. Following Yao et al. (2023), each node $v \in V$ represents a partial solution, and an edge $e \in E$ indicates that the parent's partial solution is a direct prerequisite for the child's. Let $S_{\text{leaf}}(T) \subset V$ denote the set of leaf nodes in $T$, each representing a single solution attempt, either a completed solution under one approach or a dead-end where the approach failed to yield a correct or full solution. Among these, we define $S_{\text{leaf}}^{\star}(T) = \{v \in S_{\text{leaf}}(T) : v \text{ encodes a fully correct solution}\}$ as the subset of leaves that represent correct solutions.

- **Jump layer (action)**: Let $\boldsymbol{i} = (i_0, i_1, \ldots, i_K)$ denote the sequence of reasoning steps, where $i_k$ refers to the index of $k$-th visited node in the tree. The jump starts at $v_{i_0}$ (the root) and ends at $v_{i_K}$ (the final solution). Each transition between consecutive steps $(i_k, i_{k+1})$ is labeled with an action type $\phi_k \in \{\texttt{calc}, \texttt{verify}, \texttt{backtrack}\}$, where $k = 0, \ldots, K - 1$. Here, $\texttt{calc}$ refers to generating an intermediate step via calculation or derivation; $\texttt{verify}$ denotes revisiting a node to check its correctness; and $\texttt{backtrack}$ indicates returning to a previous node to explore an alternative reasoning path. A jump is the pair $W = (\boldsymbol{i}, \boldsymbol{\phi})$, fully specifying both the node sequence and how the solver moves through the tree. A *derived solution step* is any step in the jump that reaches a leaf node via a $\texttt{calc}$ transition. Even if the leaf has been visited before, it still counts as a derived step if reached via $\texttt{calc}$; by contrast, visits for verification do not count. We mathematically define the sequence of such steps as $\boldsymbol{i}_{\text{leaf}}(T, W) = (i_{k_1}, \ldots, i_{k_M})$, where each $v_{i_{k_j}} \in S_{\text{leaf}}(T)$ and the corresponding transition is $\phi_{k_j} = \texttt{calc}$, for all $j = 1, \ldots, M$. Among these, we further define the *correct derived solution steps* as $\boldsymbol{i}_{\text{leaf}}^{\star}(T, W)$, a subsequence of $\boldsymbol{i}_{\text{leaf}}(T, W)$, consisting of indices $i_{k_j}$ such that $v_{i_{k_j}} \in S_{\text{leaf}}^{\star}(T)$.

**Scope of the transition taxonomy.** The three transition categories form an exhaustive partition at the structural level. Each step in a sequential reasoning trace either creates a new partial solution node, which is labeled $\texttt{calc}$, or revisits an existing

node. Revisits are labeled `verify` when the model checks an existing result and `backtrack` when it resumes from that result to explore a different continuation. Content-level behaviors such as knowledge recall, brainstorming, or heuristic guessing occur inside a node and are orthogonal to this traversal-level taxonomy. Similarly, repeated cycles correspond to repeated visits to existing nodes, while dependencies between subproblems are represented through tree edges or shared ancestors.

**Partial-solution granularity.** We define a node as a substantive intermediate subproblem with a clear input and output, analogous to a lemma in a proof. Minor algebraic manipulations, substitutions, setup text, and narrative strategy descriptions are merged into the surrounding substantive node. The task-specific extraction prompts in Sec. B.1 make this criterion explicit for both MATH-500 and Game of 24.

**Notations.** Define $\mathrm{consec}(\cdot)$ as an operator that takes a sequence as input and returns the set of all consecutive pairs; that is, for a sequence $(x_1, \ldots, x_M)$, $\mathrm{consec}(x_1, \ldots, x_M) = \{(x_1, x_2), \ldots, (x_{M-1}, x_M)\}$. Let $\mathrm{set}(\cdot)$ be the operator that converts a sequence into a set. For any sequence $\boldsymbol{i} = (i_1, \ldots, i_n)$, we denote its $j$-th element (1-based indexing) by $\boldsymbol{i}[j]$. When considering $N$ generated reasonings and their corresponding ReJump, we use subscript $(n)$ to denote the $n$-th reasoning instance.

Next, we present more rigorous definitions of the evaluation metrics.

### A.1. Extension of Sec. 3.2: Evaluation Metrics

This representation enables analysis of LLM reasoning behaviors, including solution diversity, exploration-exploitation trade-off, effectiveness in identifying correct paths, frequency of overthinking, forgetting, and verification. These aspects are quantified using the following metrics, computed across all reasonings and their corresponding ReJumps within a task.

**Solution Count ($\#_{\text{solution}}$).** This metric quantifies the model's ability to discover diverse solution attempts, measured by the number of leaf nodes in the reasoning tree. We define $\#_{\text{solution}}(\{T\}) = |S_{\text{leaf}}(T)|$ as the total number of leaf nodes representing distinct solutions within a single tree $T$. To assess the average performance over $N$ reasoning instances for a given task, we calculate the Average Solution Count as the mean number of unique solutions found across all instances: $\#_{\text{solution}}(\{T^{(n)}\}_{n=1}^N) = \sum_{n=1}^N \#_{\text{solution}}(\{T^{(n)}\})/N$.

**Jump Distance ($d_{\text{jump}}$).** This metric captures the exploration-exploitation tradeoff of the reasonings by averaging how far the reasoning "jumps" between newly visited leaf nodes (see Fig. 2). Define $d(u, v)$ as the number of edges on the path between nodes $u, v \in V$. The jump distance of a single jump trace is $d_{\text{jump}}(\{(T, W)\}) = \frac{1}{|\mathrm{consec}(\boldsymbol{i}_{\text{leaf}}(T,W))|} \sum_{(i_j, i_l) \in \mathrm{consec}(\boldsymbol{i}_{\text{leaf}}(T,W))} d(v_{i_j}, v_{i_l})$, and the task-level average is $d_{\text{jump}}(\{(T^{(n)}, W^{(n)})\}_{n=1}^N) = \frac{1}{N} \sum_{n=1}^N d_{\text{jump}}(\{(T^{(n)}, W^{(n)})\})$. As a robustness check, replacing the mean with the median produces nearly identical cross-model rankings: the Spearman rank correlation between the two aggregation choices is $0.95$ on both MATH-500 and Game of 24.

**Success Rate ($r_{\text{success}}$).** The metric measures how frequently a reasoning path produces a correct solution. For a single reasoning with tree $T$ and jump $W$, the success rate is computed as $r_{\text{success}}(\{(T, W)\}) = |\boldsymbol{i}_{\text{leaf}}^\star(T, W)|/|\boldsymbol{i}_{\text{leaf}}(T, W)|$, i.e., the proportion of newly visited leaf nodes that are correct solutions. The overall average is computed across all $N$ reasoning instances: $r_{\text{success}}(\{(T^{(n)}, W^{(n)})\}_{n=1}^N) = \frac{1}{N} \sum_{n=1}^N r_{\text{success}}(\{(T^{(n)}, W^{(n)})\})$.

**Verification Rate ($r_{\text{verify}}$).** This metric quantifies how frequently the model invokes verification steps during its reasoning process. For a given reasoning instance with jump $W = (\boldsymbol{i}, \boldsymbol{\phi})$, the verification rate is computed as the number of `verify` transitions divided by the total number of steps in the jump: $r_{\text{verify}}(\{W\}) = \sum_{\phi_k \in \mathrm{set}(\boldsymbol{\phi})} \mathbb{I}\{\phi_k = \texttt{verify}\}/(K-1)$. We report the average verification rate across all $N$ reasoning instances. $r_{\text{verify}}(\{W^{(n)}\}_{n=1}^N) = \sum_{n=1}^N r_{\text{verify}}(\{W^{(n)}\})/N$.

**Overthinking Rate ($r_{\text{overthinking}}$).** This metric quantifies the extent of unnecessary exploration after a correct solution has already been found. For a given reasoning instance with tree $T$ and jump $W$, let $k_0^\star$ denote the first index in $\boldsymbol{i}_{\text{leaf}}^\star(T, W)$, the step at which a correct leaf is first reached. The overthinking rate is defined as the fraction of newly visited leaf nodes that appear *after* this first correct solution: $r_{\text{overthinking}}(\{(T, W)\}) = |\{i_k \in \mathrm{set}(\boldsymbol{i}_{\text{leaf}}(T, W)) : k > k_0^\star\}|/|\boldsymbol{i}_{\text{leaf}}(T, W)|$. In other words, it measures how many additional leaf nodes are explored via `calc` transitions after a correct solution has been

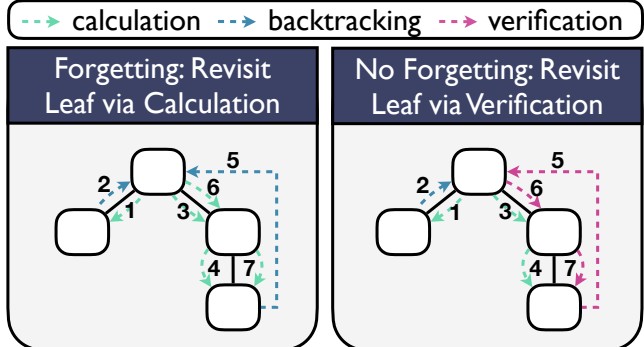

*Figure 9.* Illustration of forgetting vs. no forgetting in ReJump. Revisiting an already-seen node via `calc` indicates forgetting, while revisiting via `verify` does not.

*Table 7.* Redundancy scores of the six metrics on MATH-500 and Game of 24. Lower values indicate less dependence on other metrics.

| Dataset | $\#_{\text{solution}}$ | $d_{\text{jump}}$ | $r_{\text{success}}$ | $r_{\text{verify}}$ | $r_{\text{overthinking}}$ | $r_{\text{forget}}$ |
|---|---|---|---|---|---|---|
| MATH-500 | 0.789 | 0.624 | 0.437 | 0.105 | 0.944 | 0.277 |
| Game of 24 | 0.731 | 0.761 | 0.882 | 0.687 | 0.840 | 0.840 |

identified. Verification-only revisits are excluded because $i_{\text{leaf}}$ includes only leaf visits reached through `calc` transitions. The task-level overthinking rate is then given by the average over all $N$ reasoning instances: $r_{\text{overthinking}}(\{(T^{(n)}, W^{(n)})\}_{n=1}^{N}) = \frac{1}{N} \sum_{n=1}^{N} r_{\text{overthinking}}(\{(T^{(n)}, W^{(n)})\})$.

**Forgetting Rate ($r_{\text{forget}}$).** This metric tracks how often the model forgets its earlier steps and recomputes a previously derived result. Specifically, forgetting is flagged when a previously visited leaf node is revisited via a `calc` transition (see Fig. 9). We define a binary indicator for each reasoning instance as $\mathbf{1}_{\text{forget}}(T, W) = 1 - \prod_{m=2}^{M} \left( \prod_{j=1}^{m-1} \mathbb{I}\{i_{\text{leaf}}[j] \neq i_{\text{leaf}}[m]\} \right)$, which returns 1 if any earlier leaf is re-entered, and 0 otherwise. We use a binary instance-level indicator because forgetting events are rare; a per-step frequency would be dominated by zeros, while averaging the indicator across traces still gives a population-level rate. The forgetting rate is then reported as the proportion of instances where forgetting occurred: $r_{\text{forget}}(\{(T^{(n)}, W^{(n)})\}_{n=1}^{N}) = \frac{1}{N} \sum_{n=1}^{N} \mathbf{1}_{\text{forget}}(T^{(n)}, W^{(n)})$.

### A.2. Redundancy Analysis of Proposed Metrics

To assess whether the six proposed metrics capture complementary aspects of reasoning behavior, we conduct an information-theoretic redundancy analysis. For each metric $M$, we compute its redundancy as

$$\text{Redundancy}(M) = \frac{I(M; \text{others})}{H(M)},$$

where $H(M)$ is the entropy of $M$ and $I(M; \text{others})$ is the mutual information between $M$ and the remaining metrics. Lower values indicate that the metric contains information not recoverable from the others. Redundancy scores for MATH-500 and Game of 24 are reported in Tab. 7. The redundancy pattern is task-dependent. For example, some metrics co-vary more strongly on MATH-500, where traces are shorter and more exploitative, while Game of 24 induces more exploratory traces and changes which metrics overlap.

## B. Extended Sec. 4: ReJump-Extractor

This section provides implementation-facing details for the extraction pipeline described in Sec. 4. It includes the prompts used to construct ReJumps, examples converted from real reasoning traces, and validation studies showing that the extracted structures are stable enough for downstream analysis. In addition to the illustrative example on a math word problem, we include one more example showcasing the construction and visualization of ReJump, with both examples shown in Fig. 8.

## B.1. LLM Prompts for ReJump Construction

In this section, we present the prompt used by the LLM to parse results across all experiments.

We use different prompts for the two datasets. Listing 1 and Listing 2 show the prompts used to extract the tree and jump from the generated reasoning for MATH-500, respectively, while Listing 3 and Listing 4 show the corresponding prompts for Game of 24.

```
1 def get_tree_prompt_math(input_str, output_str):
2     return f"""
3 Your task is to analyze a detailed thinking process for solving a math problem (provided
     below) and convert it into a reasoning tree. This tree must represent the **
     chronological flow of solving substantive, mathematically well-posed subproblems or
     distinct attempts**, starting from an initial state and culminating in answering the
     original question.
4
5 Represent this structure as a **single JSON object** where keys are unique node IDs (e.g.,
      "node1", "node2") and values are node objects detailing each state or subproblem
     attempt.
6
7 **Core Principles for Tree Generation:**
8
9 * **Chronological Flow & Dependency:** The tree follows the order of substantive steps/
     attempts in the reasoning. Parent links indicate the preceding step whose `Result`
     provides necessary mathematical input.
10   **BRANCHING AND SUBSTEP RULE:**
11    - Create a new branch **if and only if** the reasoning process explicitly abandons or
     gives up on a previous approach and then starts a new, distinct solution plan. In
     other words, a new branch is created always and only when the previous line of
     reasoning is abandoned and a fundamentally different method is attempted. The new
     branch should start from the most recent shared node. Even if the solver does not
     immediately abandon the previous approach, we still consider it an Abandoned Attempt
     Node and mark it with [Path abandoned] if a different method is initiated that departs
      from the original direction.
12    - Importantly, whenever a new branch is created, the leaf node where the previous
     method ended must be explicitly marked with [Path abandoned].
13    - Conversely, if the current node is marked with [Path abandoned], a new branch must
     always be created.
14    - Importantly, for all subproblems or calculations within a single uninterrupted
     attempt, even if subcalculations are mathematically independent, represent these steps
      sequentially in the order they are performed in the reasoning: each node's parent
     must be the immediately preceding node within that attempt.
15    That is, substeps within any one attempt always form a single chain.
16 * **Substantive, Well-Posed Steps Only:** Nodes must represent **major** intermediate
     calculations or logical deductions constituting a clear, self-contained mathematical
     task (like a homework sub-problem). **Aggressively filter out** setup actions,
     strategy descriptions, narrative, verification, and trivial calculations/manipulations
     . Minor algebraic steps within a larger logical step must be grouped.
17 * **Include Failed Attempts:** Represent distinct, substantive calculation or derivation
     attempts that were **explicitly abandoned** in the reasoning as separate nodes in the
     chronological flow. **Do not filter these out.**
18 * **Focus on Mathematical Task:** Intermediate `Problem` fields must state a clear
     mathematical objective based on **all necessary given mathematical conditions and
     inputs**, avoiding descriptions of the reasoner's process or assumptions *within the
     Problem text*.
19 * **Special Final Node:** The node performing the last calculation for the final answer
     uses the original problem statement as its `Problem`.
20
21 **Node Object Structure:**
22 Each node object must contain: `Problem`, `parent`, `Result`.
23
24 1.  **`Problem` (String): Defines the specific mathematical task for this node.**
25     * **`node1` (Root):** Must be exactly "Initial State".
26     * **Intermediate Nodes (`node2` to `node(N-1)`):** Formulates a **clear,
     mathematically well-posed, and self-contained task representing a substantive step or
```

distinct attempt.** Each node represents achieving a distinct intermediate objective through calculation or deduction.
27      * **Format:** Start with "Given..." listing **all essential mathematical conditions, constraints, equations, and input values** (often from parent `Result` or established context like 'point P is on curve C') needed to define and solve *this specific task*. End with a specific mathematical question/instruction (e.g., "Calculate...", "Solve...", "Derive...").
28      * **Content:** The formulation must focus purely on the **mathematical task**, making it **understandable and solvable in isolation** like a homework sub-problem, using only the provided "Given..." information and general mathematical knowledge. **CRITICAL RULE:** The `Problem` text **must not** include descriptions of the reasoner's strategy, assumptions, or procedural instructions reflecting the reasoning flow. State only the necessary mathematical conditions and the objective. The task must be **substantive**. **CRITICAL FILTERING RULE:** **DO NOT** create separate nodes for individual algebraic manipulations... [rest of filtering rule stays the same – GROUP minor operations]. Also filter out narrative, setup, verification. No meta-tags or node ID references.
29      * **`nodeN` (Final Calculation Node):** **This node represents the very last calculation step that produces the final answer.** Its `Problem` field **must contain the verbatim Original Problem Statement.**
30
31 2.   **`parent` (String): Identifies the immediately preceding substantive step providing necessary input.**
32      * **`node1`:** Must be "none".
33      * **Other Nodes (`node2` to `nodeN`):** Must be the ID of the node whose `Result` provides the direct mathematical prerequisite for the task in the current node's `Problem`. (For abandoned attempts, the parent is the node preceding the attempt).
34
35 3.   **`Result` (String): Records the mathematical outcome of completing the task.**
36      * **`node1`:** "Original problem statement provided **as** context." (or similar).
37      * **Intermediate Nodes (`node2` to `node(N-1)`):** The direct mathematical outcome of achieving the task defined in `Problem`. Summarizes the result of grouped operations.
38      * **Abandoned Attempt Nodes:** Must state any partial outcome and explicitly end with "[Path abandoned]".
39      * **`nodeN` (Final Calculation Node):** Must be the **final answer** to the Original Problem Statement.
40
41 **Instructions for Analysis:**
42 1.   **Inputs:** Use the "Original Problem Statement" and "Input Reasoning Process".
43 2.   **Identify & Filter Steps:** Read the reasoning chronologically. Identify **major** calculation phases, key logical deductions, or distinct attempts. **Crucially, ensure that distinct, substantive attempts explicitly marked as abandoned in the reasoning are identified and *not* filtered out.** Apply the **CRITICAL FILTERING and GROUPING RULES** aggressively: Group sequences of trivial algebraic steps into the single larger objective they serve. Filter out non-mathematical content, setup, strategy descriptions/assumptions-as-actions, and verification. Only create nodes for the remaining substantive steps and distinct abandoned attempts.
44 3.   **Create Nodes Sequentially:**
45      * Create `node1`.
46      * For each identified **substantive step/objective/attempt** *before* the final answer calculation: Create the corresponding intermediate node (`node2`, `node3`, ...). Determine `parent`. Formulate the `Problem` strictly according to Rule 1 (well-posed, self-contained task including **all necessary conditions/constraints**, no process descriptions). Record `Result`. Link abandoned attempt nodes chronologically.
47      * For the **final calculation step**: Create `nodeN`. Determine `parent`. Set `Problem` to verbatim Original Problem Statement. Set `Result` to final answer.
48 4.   **Formatting:** Use LaTeX (`$...$`) for all math notation.
49 5.   **Output:** Produce a single JSON object.
50
51 ---
52 **BEGIN ORIGINAL PROBLEM STATEMENT**
53 ---
54 {input_str}
55 ---
56 **END ORIGINAL PROBLEM STATEMENT**

```
57 ---
58
59 ---
60 **BEGIN INPUT REASONING PROCESS**
61 ---
62 {output_str}
63 ---
64 **END INPUT REASONING PROCESS**
65 ---
66
67 Generate the JSON output based on these instructions.
68     """
69
70 # After obtaining the tree, use a separate prompt to evaluate the correctness of each leaf
        node for refining the tree.
71 def get_result_parsing_and_comparison_prompt(result_string, ground_truth_string):
72     return f"""You are an expert AI assistant. Your task is to analyze a 'Result' string
    from a mathematical reasoning step and compare its final numerical answer to a 'Ground
     Truth' value.
73
74 Instructions:
75 1.  Extract the final numerical value(s) from the 'Result' string.
76     - If multiple numbers are present, focus on the one that seems to be the conclusive
    answer of that step.
77     - Handle approximations (e.g., "approx 46.0", "is about 3.14").
78     - If the result explicitly states abandonment (e.g., "[Path abandoned]"), extract the
    numerical value derived *before* abandonment, if any. If no clear numerical value was
    derived, use "N/A" for the parsed value.
79     - If no specific numerical answer can be clearly identified, use "N/A" for the parsed
    value.
80
81 2.  Compare the extracted numerical value with the 'Ground Truth' value.
82     - The comparison should determine if they are essentially the same, considering
    potential minor differences in formatting or precision (e.g., "46" vs "46.0", "1.03"
    vs "1.035" if context implies rounding).
83     - If the parsed value is "N/A", the comparison result should be "NOT_APPLICABLE".
84     - If the ground truth is empty or clearly not a comparable numerical value, and the
    parsed value is numerical, consider it a "MISMATCH" unless specified otherwise.
85
86 3.  Output a single JSON object with two keys:
87     -   `"parsed_value"`: The extracted numerical value as a string (e.g., "46", "3.14", "
    N/A").
88     -   `"match_status"`: A string indicating the comparison result. Must be one of: "
    MATCH", "MISMATCH", "NOT_APPLICABLE".
89
90 Example:
91 Result string: "Using the approximations, $tan x^\circ \\approx \\frac{{1.3270 +
    6.3138}}{{1.3270 \\times 6.3138 - 1}} \\approx \\frac{{7.6408}}{{8.381 - 1}} \\approx
    \\frac{{7.6408}}{{7.381}} \\approx 1.0355$. This implies $x \\approx arctan(1.0355) \\
    approx 46.0^\circ$. [Path abandoned]"
92 Ground Truth string: "46"
93 Expected JSON Output: {{"parsed_value": "46.0", "match_status": "MATCH"}}
94
95 Result string: "The answer is $y=3$."
96 Ground Truth string: "3.0"
97 Expected JSON Output: {{"parsed_value": "3", "match_status": "MATCH"}}
98
99 Result string: "The calculation leads to $10/2 = 5$. However, this path is incorrect."
100 Ground Truth string: "7"
101 Expected JSON Output: {{"parsed_value": "5", "match_status": "MISMATCH"}}
102
103 Result string: "[Path abandoned] No value obtained."
104 Ground Truth string: "10"
105 Expected JSON Output: {{"parsed_value": "N/A", "match_status": "NOT_APPLICABLE"}}
106
```

```
107  ---
108  Result string to analyze:
109  {result_string}
110
111  Ground Truth value:
112  {ground_truth_string}
113  ---
114
115  JSON Output:"""
```

*Listing 1.* Prompt for extracting a tree from the reasoning trace in JSON format for MATH-500.

```
1  def get_jump_prompt(input_str, output_str, tree_json):
2      return f"""
3  You are an AI assistant specialized in analyzing mathematical reasoning processes. Your
       task is to trace the provided reasoning text against a structured reasoning tree and
       generate a "walk" representing the trajectory of the thought process.
4
5  **Inputs:**
6
7  1.  **Problem Description:**
8      ```
9      {input_str}
10     ```
11 2.  **Reasoning Text:** A step-by-step textual explanation of how the problem was solved,
       including potential errors, corrections, explorations of different paths, and
       verifications.
12     ```text
13     {output_str}
14     ```
15 3.  **Reasoning Tree:** A JSON object representing the structured steps and dependencies
       of the solution(s). Each key is a node ID, and the value contains information about
       that step, including its parent node and specifically a "Problem" field describing the
        task of that node.
16     ```json
17     {tree_json}
18     ```
19
20 **Task:**
21
22 Analyze the `Reasoning Text` to determine the sequence in which the solver mentally
       visited or considered the steps represented by the nodes in the `Reasoning Tree`.
       Identify the transitions between these nodes and categorize each transition. **
       Crucially, for verification steps, visiting a node X implies the text shows evidence
       of re-doing the specific task described in the "Problem" field of node X.**
23
24 **Output Format:**
25
26 Generate a JSON list of dictionaries, where each dictionary represents a single step in
       the reasoning walk. Each dictionary must have the following keys:
27
28 * `from`: The ID (string) of the node the reasoning is moving *from*.
29 * `to`: The ID (string) of the node the reasoning is moving *to*.
30 * `category`: A string indicating the type of transition. Must be one of:
31     * `calculation/derivation`: Represents forward progress in the reasoning, moving from
       one step to the next logical step (often parent to child in the tree) to derive new
       information or explore a solution path.
32     * `backtracking`: Represents abandoning a current line of thought or calculation (
       often because it's incorrect, inefficient, or a dead end) and returning to a previous
       state (node) to try a different approach. This is typically a move from a node to one
       of its ancestors (not necessarily the direct parent).
33     * `verification`: Represents checking or confirming a result or step **by re-doing the
        work associated with previous nodes**. This is determined based on the text:
34         * **Specific Re-work:** If the text explicitly describes actions that precisely
       match the **problem description** defined within an intermediate node (e.g., node X)
```

as part of checking a later result (node Z), trace the path reflecting that specific re-work (e.g., Z -> X -> Z). This requires clear evidence in the text of **re-solving the problem defined in node X**.

35     * **General Check:** If the text indicates verification of a result (node Z) but ***does not*** show actions matching the specific **problem description** of any intermediate node, interpret this as checking consistency with the initial problem statement/conditions (node 1). Represent this path as Z -> 1 -> Z. ***Note: Simply using a formula or result from a previous node (e.g., node X) without showing the steps to re-solve the problem defined in node X does NOT count as re-doing the work of node X.***

36

37 **Instructions:**

38

39 1.  Read the `Reasoning Text` carefully, paying attention to the flow, changes in direction, calculations, statements of intent (e.g., "Let me **try**...", "No, that's wrong...", "Let me verify..."), and results.
40 2.  Map segments of the `Reasoning Text` to the corresponding nodes in the `Reasoning Tree`. Use the "Problem" and "Result" fields in the tree nodes to help with mapping *initial* derivations.
41 3.  Identify the sequence of nodes visited or considered based on the flow of the `Reasoning Text`.
42 4.  For each transition from one node (`from`) to the next (`to`) in the sequence, determine the appropriate `category` based the definitions above.
43 5.  Pay close attention to parts of the reasoning text that indicate:
44     * Starting a calculation or derivation (maps to `calculation/derivation`).
45     * Realizing an error or deciding a path is not fruitful and returning to an earlier idea (maps to `backtracking`).
46     * Re-checking results (maps to `verification`). **When mapping `verification`:** First, check if the text describes actions that precisely match the **problem description** of an intermediate node (Node X), essentially re-doing the work defined in that node. If yes, trace the walk through the node being re-worked (e.g., Z -> X -> Z). If the text indicates verification but ***does not*** show such a specific re-work of a prior node's problem, assume it implies checking against the initial problem conditions (node 1) **and** represent the path **as** Z -> 1 -> Z. Remember: Simply *using* a result **or** formula **from** node X does **not** qualify **as** re-doing the problem of node X according to this definition.
47 6.  The walk should reflect the *actual* path taken **in** the `Reasoning Text`, including explorations of dead ends (like `node2` **in** the example) **and** subsequent backtracking.

48

49     **Mandatory Backtracking Rule:**
50     Only when the reasoning process explicitly abandons **or** gives up on the current approach at node A **and** then starts a new, distinct attempt at node B must you include a backtracking transition **from** A to the parent of B, followed by a calculation/derivation transition **from** the parent of B to B. Never allow a direct calculation/derivation transition **from** A to B **in** these cases. Do **not** include backtracking transitions **except in** such abandonment cases.

51

52 7.  Ensure the output **is** strictly the JSON **list as** specified, **with** no additional explanatory text.
53 8. The output MUST be perfectly valid JSON, parseable by standard libraries.
54 9. The walk must always start at node1: The first transition **in** your output should always be `"from": "node1"`, `"to": ...`. Never use `"from": "none"`, `"from": null`, **or any** other alternative. Assume reasoning always conceptually begins at node1.

55

56 **Example Analysis (Based on Provided Inputs **with** Stricter Verification Logic):**

57

58 * Reasoning starts, defining the problem (maps to `node1`).
59 * Text explores calculating AB **with** specific points (maps to `node2`). `node1` -> `node2` (`calculation/derivation`).
60 * Text says "That seems messy... Let me think differently." **and** abandons the `node2` approach, returning to the setup phase (conceptually `node1`). `node2` -> `node1` (`backtracking`).
61 * Text introduces symmetry **and** points B(x,y), C(-x,y) (maps to `node3`). `node1` -> `node3` (`calculation/derivation`). This step involves *doing* the problem **in** `node3` (calculating distances).

```
62 * Text derives relationship between AB and BC, sets them equal (maps to `node4`). `node3`
      -> `node4` (`calculation/derivation`).
63 * Text solves for x and y using parabola equation (maps to `node5`). `node4` -> `node5` (`
      calculation/derivation`).
64 * Text calculates final side length (maps to `node6`). `node5` -> `node6` (`calculation/
      derivation`).
65 * Text says "Let me verify with the distance." It then shows:
66    1.  `AB = sqrt(x^2 + y^2) = ...` This ***uses*** the formula derived in `node3` and
      values from `node5`. It does ***not*** show a re-derivation of the distance formula as
       described in `node3`'s problem ("Calculate the distances...").
67    2.  `BC is 2x = ...` This ***uses*** the formula derived in `node3` and value from `
      node5`. It does ***not*** show a re-derivation.
68 * **Applying the strict verification rule:** Does the text show actions matching the *
      problem description* of an intermediate node (like re-deriving the formulas as defined
       in `node3`'s problem, or re-solving for x,y as defined in `node5`'s problem)? **No**,
       the text only shows the *application* of results from previous nodes.
69 * Therefore, according to the rule, since no specific re-work of a prior node's **problem
      ** is detailed, we default to the **General Check** case. The path should be
      represented as checking the final result (`node6`) against the initial state (`node1`)
      .
70 * The expected verification path for this text, under this strict interpretation, would be
      : `node6` -> `node1` (`verification`), potentially followed by `node1` -> `node6` (`
      verification`) or repeated. A simple `node6 -> node1 -> node6` sequence for the
      overall verification check is likely.
71
72 **Final Output Request:**
73
74 Now, analyze the provided inputs (`{{problem_description}}`, `{{reasoning_text}}`, `{{
      reasoning_tree_json}}`) using **this strict interpretation of verification** (visiting
       a node requires re-doing its specific "Problem") and generate the reasoning walk as a
       JSON list. Output *only* the JSON list.
75    """
```

*Listing 2.* Prompt used to extract jump from reasoning as a JSON structure for MATH-500.

```
1 def get_tree_prompt(input_str, output_str):
2     return f"""
3 Given the problem statement and reasoning process below. Your task is to analyze a
      detailed thinking process for solving a math problem (provided below) and convert it
      into a reasoning tree. **Do not try to solve the problem yourself, fully use the given
       reasoning process and just convert it!**
4
5 ---
6 **BEGIN ORIGINAL PROBLEM STATEMENT**
7 ---
8 {input_str}
9 ---
10 **END ORIGINAL PROBLEM STATEMENT**
11 ---
12
13 ---
14 **BEGIN INPUT REASONING PROCESS**
15 ---
16 {output_str}
17 ---
18 **END INPUT REASONING PROCESS**
19 ---
20
21 Here are some instructions:
22
23 **Node Object Structure:**
24
25 Each node object must contain: `Problem`, `parent`, `Result`.
26
27 1. **`Problem` (String): A partial solution containing the four numbers and any
```

```
          calculation has been tried. Only use numbers, + - * / and parentheses.
28
29 * **`node1` (Root):** Must be exactly the four initial numbers in the problem. For example
       , "9,3,12,8".
30
31 * **Non-leaf Nodes:** Each node describes the partial solution being explored. For example
       , for problem 9,3,12,8, an intermediate node "9-3, 12, 8" means that we have tried
       (9-3), and need to try 2 more calculations with numbers 12 and 8 to get 24. Give all
       these nodes indexes number to keep tracking (after node1).
32
33 * **Leaf node:** **This node represents the very last calculation step that produces the
       final answer after three calculation steps.** For example, for problem 9,3,12,8, this
       could be "9-3+128", which is a leaf node that is unsuccessful. Another successful leaf
        node could be "(9-3)*(128)". Also use an index number for each one (after node1).
34
35 Pay attention that the problem statement of each node should be unique. If two nodes have
       the same description (i.e., the same partial calculation and the numbers not
       calculated so far), merge them into one.
36
37 2. **`parent` (String):
38
39 * **`node1` (root):** Must be None.
40
41 * **Other nodes:** Must be the previous partial solution that the current node builds on.
       For example, the parent of the node "9-3, 12, 8" is "9,3,12,8". But here just use the
       index number to indicate the index of its parent node.
42
43 3. **`Result` (String):
44
45 * **`root`:** None.
46
47 * **Intermediate Nodes:** None.
48
49 * **Leaf node** Must be the **final answer**. For example, the result of node "9-3+12-8"
       is 10. Written in latex.
50
51 Please generate a single JSON output. This output must be a **single JSON object** where
       keys are unique node IDs (e.g., "node1", "node2", corresponding to the index numbers
       assigned to track the nodes) and values are the node objects (containing 'Problem', '
       parent', 'Result') as detailed above.
52
53     """
```

*Listing 3.* Prompt used to extract tree from reasoning as a JSON structure for Game of 24.

```
1 def get_jump_prompt(input_str, output_str, tree_json):
2     return f"""
3 You are an AI assistant specialized in analyzing mathematical reasoning processes. Your
       task is to trace the provided reasoning text against a structured reasoning tree and
       generate a "walk" representing the trajectory of the thought process.
4
5 **Inputs:**
6
7 1.  **Problem Description:**
8     ```
9     {input_str}
10    ```
11 2.  **Reasoning Text:** A step-by-step textual explanation of how the problem was solved,
       including potential errors, corrections, explorations of different paths, and
       verifications.
12    ```text
13    {output_str}
14    ```
15 3.  **Reasoning Tree:** A JSON object representing the structured steps and dependencies
       of the solution(s). Each key is a node ID, and the value contains information about
```

```
          that step, including its parent node and specifically a "Problem" field describing the
           task of that node.
16      ```json
17      {tree_json}
18      ```
19
20  **Task:**
21
22  Analyze the `Reasoning Text` to determine the sequence in which the solver mentally
        visited or considered the steps represented by the nodes in the `Reasoning Tree`.
        Identify the transitions between these nodes and categorize each transition. **
        Crucially, for verification steps, visiting a node X implies the text shows evidence
        of re-doing the specific task described in the "Problem" field of node X.**
23
24  **Output Format:**
25
26  Generate a JSON list of dictionaries, where each dictionary represents a single step in
        the reasoning walk. Each dictionary must have the following keys:
27
28  * `from`: The ID (string) of the node the reasoning is moving *from*.
29  * `to`: The ID (string) of the node the reasoning is moving *to*.
30  * `category`: A string indicating the type of transition. Must be one of:
31      * `calculation/derivation`: Represents forward progress in the reasoning, moving from
        one step to the next logical step (often parent to child in the tree) to derive new
        information or explore a solution path.
32      * `backtracking`: Represents abandoning a current line of thought or calculation (
        often because it's incorrect, inefficient, or a dead end) and returning to a previous
        state (node) to try a different approach. This is typically a move from a node to one
        of its ancestors (not necessarily the direct parent).
33      * `verification`: Represents checking or confirming a result or step **by re-doing the
         work associated with previous nodes**. This is determined based on the text:
34          * **Specific Re-work:** If the text explicitly describes actions that precisely
        match the **problem description** defined within an intermediate node (e.g., node X)
        as part of checking a later result (node Z), trace the path reflecting that specific
        re-work (e.g., Z -> X -> Z). This requires clear evidence in the text of **re-solving
        the problem defined in node X**.
35          * **General Check:** If the text indicates verification of a result (node Z) but
        ***does not*** show actions matching the specific **problem description** of any
        intermediate node, interpret this as checking consistency with the initial problem
        statement/conditions (node 1). Represent this path as Z -> 1 -> Z. ***Note: Simply
        using a formula or result from a previous node (e.g., node X) without showing the
        steps to re-solve the problem defined in node X does NOT count as re-doing the work of
         node X.***
36
37  **Instructions:**
38
39  1.  Read the `Reasoning Text` carefully, paying attention to the flow, changes in
        direction, calculations, statements of intent (e.g., "Let me **try**...", "No, that's
        wrong...", "Let me verify..."), and results.
40  2.  Map segments of the `Reasoning Text` to the corresponding nodes in the `Reasoning Tree
        `. Use the "Problem" and "Result" fields in the tree nodes to help with mapping *
        initial* derivations.
41  3.  Identify the sequence of nodes visited or considered based on the flow of the `
        Reasoning Text`.
42  4.  For each transition from one node (`from`) to the next (`to`) in the sequence,
        determine the appropriate `category` based the definitions above.
43  5.  Pay close attention to parts of the reasoning text that indicate:
44      * Starting a calculation or derivation (maps to `calculation/derivation`).
45      * Realizing an error or deciding a path is not fruitful and returning to an earlier
        idea (maps to `backtracking`).
46      * Re-checking results (maps to `verification`). **When mapping `verification`:** First
        , check if the text describes actions that precisely match the **problem description**
         of an intermediate node (Node X), essentially re-doing the work defined in that node.
         If yes, trace the walk through the node being re-worked (e.g., Z -> X -> Z). If the
        text indicates verification but ***does not*** show such a specific re-work of a prior
```

node's problem, assume it implies checking against the initial problem conditions (node 1) **and** represent the path **as** Z -> 1 -> Z. Remember: Simply *using* a result **or** formula **from** node X does **not** qualify **as** re-doing the problem of node X according to this definition.

47 6.  The walk should reflect the *actual* path taken **in** the `Reasoning Text`, including explorations of dead ends (like `node2` **in** the example) **and** subsequent backtracking.

48

49    **Mandatory Backtracking Rule:**
50    Only when the reasoning process explicitly abandons **or** gives up on the current approach at node A **and** then starts a new, distinct attempt at node B must you include a backtracking transition **from** A to the parent of B, followed by a calculation/ derivation transition **from** the parent of B to B. Never allow a direct calculation/ derivation transition **from** A to B **in** these cases. Do **not** include backtracking transitions **except in** such abandonment cases.

51

52 7.  Ensure the output **is** strictly the JSON **list as** specified, **with** no additional explanatory text.
53 8. The output MUST be perfectly valid JSON, parseable by standard libraries.
54 9. The walk must always start at node1: The first transition **in** your output should always be `"from": "node1"`, `"to": ...`. Never use `"from": "none"`, `"from": null`, **or any** other alternative. Assume reasoning always conceptually begins at node1.

55

56 **Example Analysis (Based on Provided Inputs with Stricter Verification Logic):**

57

58 * Reasoning starts, defining the problem (maps to `node1`).
59 * Text explores calculating AB **with** specific points (maps to `node2`). `node1` -> `node2` (`calculation/derivation`).
60 * Text says "That seems messy... Let me think differently." **and** abandons the `node2` approach, returning to the setup phase (conceptually `node1`). `node2` -> `node1` (`backtracking`).
61 * Text introduces symmetry **and** points B(x,y), C(-x,y) (maps to `node3`). `node1` -> `node3` (`calculation/derivation`). This step involves *doing* the problem **in** `node3` (calculating distances).
62 * Text derives relationship between AB **and** BC, sets them equal (maps to `node4`). `node3` -> `node4` (`calculation/derivation`).
63 * Text solves **for** x **and** y using parabola equation (maps to `node5`). `node4` -> `node5` (`calculation/derivation`).
64 * Text calculates final side length (maps to `node6`). `node5` -> `node6` (`calculation/ derivation`).
65 * Text says "Let me verify with the distance." It then shows:
66    1.  `AB = sqrt(x^2 + y^2) = ...` This ***uses*** the formula derived **in** `node3` **and** values **from** `node5`. It does ***not*** show a re-derivation of the distance formula **as** described **in** `node3`'s problem ("Calculate the distances...").
67    2.  `BC is 2x = ...` This ***uses*** the formula derived in `node3` and value from `node5`. It does ***not*** show a re-derivation.
68 * **Applying the strict verification rule:** Does the text show actions matching the *problem description* of an intermediate node (like re-deriving the formulas as defined in `node3`'s problem, **or** re-solving **for** x,y **as** defined **in** `node5`'s problem)? **No**, the text only shows the *application* of results from previous nodes.
69 * Therefore, according to the rule, since no specific re-work of a prior node's **problem** **is** detailed, we default to the **General Check** case. The path should be represented **as** checking the final result (`node6`) against the initial state (`node1`).
70 * The expected verification path **for** this text, under this strict interpretation, would be: `node6` -> `node1` (`verification`), potentially followed by `node1` -> `node6` (`verification`) **or** repeated. A simple `node6 -> node1 -> node6` sequence **for** the overall verification check **is** likely.

71

72 **Final Output Request:**

73

74 Now, analyze the provided inputs (`{{problem_description}}`, `{{reasoning_text}}`, `{{reasoning_tree_json}}`) using **this strict interpretation of verification** (visiting a node requires re-doing its specific "Problem") **and** generate the reasoning walk **as** a JSON **list**. Output *only* the JSON **list**.

```
75      """
```

*Listing 4.* Prompt used to extract jump from reasoning as a JSON structure for Game of 24.

### B.2. Example ReJump Representations Converted from Real Reasoning Traces

In this part, we present example ReJump representations derived from real reasoning traces on both the MATH-500 and Game of 24 tasks. Specifically, we randomly select one problem from each task:

- **MATH-500**: The operation $\&$ is defined for positive integers $a$ and $b$ as $a\&b = \dfrac{\sqrt{ab+a}}{\sqrt{ab-b}}$. What is the value of $9\&2$? Express your answer as a common fraction in simplest radical form.
- **Game of 24**: Given the numbers 2, 8, 10, and 10, use each exactly once with basic arithmetic operations $(+, -, \times, \div)$ and optional parentheses to obtain 24.

For MATH500, we present the reasoning traces generated by Grok 3 Mini Beta and DeepSeek-R1 in Listing 5 and Listing 6, together with their corresponding ReJump representations shown in Fig. 11a and Fig. 11b. For Game of 24, we include the reasoning traces produced by DeepSeek-R1, Phi-4-reasoning-plus, and Claude 3.7 Sonnet in Listing 7, Listing 8, and Listing 9, along with their respective ReJump representations visualized in Fig. 10a, Fig. 10b, and Fig. 10c.

```
1 First, the operation is defined as \( a \& b = \frac{\sqrt{a b + a}}{\sqrt{a b - b}} \). I
      need to find the value for \( a = 9 \) and \( b = 2 \).
2
3 So, plugging in the values: \( 9 \& 2 = \frac{\sqrt{9 \cdot 2 + 9}}{\sqrt{9 \cdot 2 - 2}}
      \).
4
5 Let me calculate inside the square roots.
6
7 First, \( 9 \cdot 2 = 18 \).
8
9 So, numerator: \( \sqrt{18 + 9} = \sqrt{27} \).
10
11 Denominator: \( \sqrt{18 - 2} = \sqrt{16} \).
12
13 Okay, so \( 9 \& 2 = \frac{\sqrt{27}}{\sqrt{16}} \).
14
15 I know that \( \frac{\sqrt{a}}{\sqrt{b}} = \sqrt{\frac{a}{b}} \), so this simplifies to \(
      \sqrt{\frac{27}{16}} \).
16
17 Now, \( \frac{27}{16} \) is already a fraction, and 27 is 9 times 3, which is 3 squared
      times 3, so \( \sqrt{27} = \sqrt{9 \cdot 3} = 3\sqrt{3} \).
18
19 And \( \sqrt{16} = 4 \).
20
21 So, originally, \( \frac{\sqrt{27}}{\sqrt{16}} = \frac{3\sqrt{3}}{4} \).
22
23 Is that the simplest form? Yeah, 3 and 4 have no common factors, and   3  is already
      simplified.
24
25 The problem says "express your answer as a common fraction in simplest radical form." So,
      \( \frac{3\sqrt{3}}{4} \) is a common way to write it.
26
27 Is this a fraction? It's a fractional expression with a radical, which I think qualifies.
28
29 I could write it as \( \frac{3}{4} \sqrt{3} \), but that's the same thing.
30
31 Let me double-check my calculations.
32
33 a = 9, b = 2
34
35 ab + a = 9*2 + 9 = 18 + 9 = 27,   2 7 =     (9*3) = 3   3
36
37 ab - b = 18 - 2 = 16,    1 6 = 4
```

```
38
39 So, yes, \( \frac{3\sqrt{3}}{4} \)
40
41 I think that's the final answer.
42
43 The problem says "Your response should just be the answer containing only answer with no
      additional text for example, 0"
44
45 So, I should just write \( \frac{3\sqrt{3}}{4} \)
46
47 In the example, it's just "0", so probably as a mathematical expression.
48
49 Sometimes they write it as a fraction.
50
51 I think it's fine.
52
53 One more thing: is there any simplification I missed? Is the expression inside the square
      roots positive? For a=9, b=2, ab + a = 27 > 0, ab - b = 16 > 0, so yes.
54
55 Also, b is positive integer, as stated.
56
57 Alright, I think that's it.
```

*Listing 5.* Example reasoning generated by Grok 3 Mini Beta on MATH-500.

```
1 Okay, let's try to solve this problem step by step. The operation is defined as a & b =
      sqrt(ab + a) / sqrt(ab - b). We need to find the value of 9 & 2. Alright, so first, I
      should substitute a = 9 and b = 2 into the formula. Let me write that down.
2
3 So, substituting, we get:
4
5 9 & 2 = sqrt(9*2 + 9) / sqrt(9*2 - 2)
6
7 Let me compute the numerator and denominator separately to avoid confusion.
8
9 Starting with the numerator: 9*2 + 9. Let's calculate that. 9*2 is 18, and then adding 9
      gives 18 + 9 = 27. So the numerator is sqrt(27).
10
11 Now the denominator: 9*2 - 2. That's 18 - 2 = 16. So the denominator is sqrt(16).
12
13 So now the expression becomes sqrt(27) / sqrt(16). Let me simplify each square root.
14
15 sqrt(27) can be simplified. Since 27 = 9*3, sqrt(9*3) = sqrt(9)*sqrt(3) = 3*sqrt(3).
16
17 sqrt(16) is straightforward. sqrt(16) = 4.
18
19 So substituting back, we have 3*sqrt(3) / 4. That simplifies to (3/4)*sqrt(3). But let me
      check if this is the simplest radical form.
20
21 The problem asks for a common fraction in simplest radical form. A common fraction means
      that the numerator and denominator are integers, and the radical should be simplified
      as much as possible. Here, sqrt(3) can't be simplified further, and the fraction 3/4
      is already in simplest terms. So I think that's the answer.
22
23 Wait, let me double-check my steps to make sure I didn't make any mistakes.
24
25 First, substituting a=9 and b=2:
26
27 Numerator: ab + a = 9*2 + 9 = 18 + 9 = 27. Correct.
28
29 Denominator: ab - b = 9*2 - 2 = 18 - 2 = 16. Correct.
30
31 sqrt(27) = 3*sqrt(3), sqrt(16) = 4. So 3*sqrt(3)/4. Yes, that seems right.
32
33 Hmm. Let me check if there's any other way to simplify or if I messed up the order of
```

operations. For example, in the denominator, ab − b: since it's 9*2 first, then
subtract 2, which is 18 − 2 = 16. Correct. If it had been 9*(2 − 2), that would be 0,
but that's not the case here. Parentheses are around ab in both numerator and
denominator, so the operations are multiplication first, then addition or subtraction.
So, yes, that's correct.

34

35 Another thing to check: sqrt(ab + a) / sqrt(ab − b). So the entire ab term is under the
square roots, not just parts of them. So yes, sqrt(ab +a) is sqrt((ab) + a) which is
sqrt(27), and sqrt(ab − b) is sqrt(16). All correct.

36

37 Therefore, 3*sqrt(3)/4 is the simplified form, which is a common fraction with a radical
in the numerator. Since 3 and 4 have no common factors besides 1, the fraction is
reduced completely. The radical sqrt(3) cannot be simplified further. So this should
be the answer.

38

39 Just to be thorough, let me compute the decimal values to see if it makes sense. sqrt(27)
is about 5.196, and sqrt(16) is 4. So 5.196 / 4 ≈ 1.299. If I compute 3*sqrt(3)/4,
sqrt(3) is approximately 1.732, so 3*1.732 ≈ 5.196, divided by 4 is the same 1.299.
That seems reasonable. Let me check with the original formula: sqrt(ab +a) is sqrt(27)
≈ 5 .196, sqrt(ab −b) is sqrt(16)=4, so the ratio is indeed approximately 1.299. So 3
√3 /4 is correct.

40

41 Alternatively, is there a different way to approach this problem? Maybe factoring out
terms?

42

43 Let's see. Let's take the original expression:

44

45 sqrt(ab + a) / sqrt(ab − b)

46

47 Factor numerator and denominator:

48

49 Numerator: sqrt(a(b + 1))

50

51 Denominator: sqrt(b(a − 1))

52

53 So the expression becomes sqrt(a(b + 1)) / sqrt(b(a − 1))

54

55 Which is sqrt[ (a(b + 1)) / (b(a − 1)) ) ]

56

57 But maybe that's not helpful here. Let's plug in a=9 and b=2:

58

59 sqrt(9*(2 + 1)) / sqrt(2*(9 − 1)) = sqrt(9*3) / sqrt(2*8) = sqrt(27)/sqrt(16) = same as
before. So same result. So that method also gives 3√3 /4. So yes, same answer.

60

61 Therefore, I'm confident that the value of 9 & 2 is 3√3 /4. Since the problem asks for
the answer as a common fraction in simplest radical form, and 3/4 is a common fraction
, and √3 is simplified, this should be correct.

62

63 **Final Answer**
64 \boxed{\dfrac{3\sqrt{3}}{4}}

*Listing 6.* Example reasoning generated by DeepSeek-R1 on MATH-500.

1 Okay, let's see. I need to make 24 using the numbers 2, 8, 10, and 10. Each number exactly
once.
2 Hmm. Let's start by thinking about possible combinations.\n\n
3 First, I know that 24 can be achieved through various operations. Maybe multiplication and
division could help here.
4 Let's look at the numbers. There's two 10s, which are pretty big. Maybe subtract them to
get a smaller number?
5 Wait, 10 − 10 is 0, which might not be helpful.
6 Alternatively, 10 divided by 10 is 1. Then maybe 8*2*1 would be 16, which is too low. Not
quite there.\n\n
7 Alternatively, maybe combining 8 and 2 first. 8*2 is 16. Then what's left is 10 and 10. If
I subtract them, 10−10=0, so 16+0=16. No.

```
 8 If I add them, 10+10=20, and 16+20=36, which is over. Hmm.
 9 Maybe multiply 10 and 10? That's 100, which seems too big. T
10 hen 100 divided by something. 100 divided by (8-2) is 100/6   1 6 .666, not helpful.\n\n
11 Wait, maybe using division. Let's see. If I take 10 divided by (10 - 8), that's 10/2=5.
12 Then multiply by 2? 5*2=10. Still not enough.
13 Alternatively, 8/(10-10) is undefined, so that's no good.\n\nAnother approach: maybe 2*(10
       + 10) is 40.
14 Then 40 minus 8 is 32. Close, but not 24.
15 What if we do 2*10 + 10 -8? That's 20 +10=30-8=22. Nope.\n
16 Wait, maybe combine 8 and 10 first. 10 - (8/2) =10-4=6. Then 6 * (10 - something).
17 Wait, but I have to use all numbers once. Let's see. If I do 8 divided by 2, that's 4.
18 Then 10 -4=6. Then 10*6=60. No, that's too much. Maybe 10 - (8/2) =6, then 10-6=4. But
       that uses 10 twice and not sure.\n\n
19 Let me try a different combination. What if I do (10 * 10) - (8 * 2)? That's 100 -16=84.
       No. Not helpful. Hmm.\n\n
20 Wait, 24 can be 3*8. Let's see if I can get a 3. Maybe (10 - (10/2)) is 10-5=5. Then
       5*8=40. No. Not 24. Or 10/ (10/ (8-2)) ) =10/(10/6)=10*(6/10)=6. Then 6* something.
       But maybe not.\n\n
21 Alternatively, 8 * (2 + (10/10)). Let's see: 10/10 is 1, plus 2 is 3. Then 8*3=24.
22 Oh! Wait, that uses all numbers: 2, 8, 10, 10.
23 Let me check:\n\nYes: 10 divided by 10 is 1. Then add 2: 2 +1=3. Multiply by 8: 8*3=24.
24 That uses all four numbers exactly once. So the equation is 8*(2 + (10/10)) =24.
25 Let me confirm:\n\n10 divided by 10 is 1. 2 plus 1 is 3. 8 times 3 is 24. Yep. That works
       !"
```

*Listing 7.* Example reasoning generated by DeepSeek-R1 on Game of 24.

```
 1 We are given a game 24 problem: 4 numbers: 2,8,10,10.
 2 We have the arithmetic operations allowed: addition, subtraction, multiplication, division.
        We want to use each number exactly once.
 3 And our answer is required to be provided in a block using <answer>and</answer> tags such
       that the answer is the final equation, containing only letters of the correct answer
       with no additional text.
 4 I need to provide the startup. The question says "your thinking process" but condition: "
       Now give your thinking process, then final answer in the provided tags".
 5 Actually the question says "Please provide one feasible solution", also "provide your
       thinking process in tags." Wait: the problem says "Your goal is to use all the 4
       numbers ...".
 6 The problem requires me to produce answer and include my thinking process inside "analysis
       " of the model, but then the final answer block, within <answer> ... everything?
 7 But then instructions: "your final answer should be enclosed in <answer> and </answer>
       tags, containing only letter of the correct answer with no additional text."
 8 And the example showing: <answer>2*9+18/3=24</answer>.\n\n
 9 We must produce final answer solution, which must be enclosed in <answer>... tags.
10 But instructions require reasoning tags "in tags".
11 I cannot output markdown formatting?
12 Actually instructions: "Your answer should be enclosed in <answer> and </answer> tags,
       containing only letter of the correct answer with no additional text."
13 This means the final answer must be in a code block with only the equation?
14 It reads: "final answer should be enclosed in <answer> and </answer> tags" perhaps
       similarly to <answer> ... </answer> with no additional text.
15 But instructions ask: "Please provide your thinking process in".
16 I must produce "explain my thinking process here". Perfect.\n\n
17 I produce answer: Use equation: (10 / (10 - 2/ (8-?))) let me solve possibilities:\nWe
       have numbers: 2,8,10,10.
18 Let me re-read the conversation: "Now given a game 24 problem, we have 4 numbers: 2, 8,
       10, and 10. \n                    Your goal is to use all the 4 numbers and basic
       arithmetic operations (+ - * /) to obtain 24.\n                    You must use each
       number exactly once, and you can use parentheses to change the order of operations.\n
                          Please provide one feasible solution to this problem.\n
             Please provide your thinking process in."\n\n
19 It doesn't specify a format though the final answer result must be with final answer
       enclosed in <answer>...</answer> tags.
20 It then says after that "After that, your final answer should be enclosed in <answer> and
       </answer> tags, containing only letter of the correct answer with no additional text".
```

21 The provided sample is "<answer>2*9+18/3=24</answer>".\n\n
22 But then they say "please provide your thinking process in tags" Actually instructions: "
   Please provide your thinking process in" It might be "explain your final answer with
   chain-of-thought except that we in answer if they expect previous with chain-of-
   thought there is one with text "analysis" in my chain-of-thought but my final answer
   should be enough to be valid.
23 The instructions require my output string to have one final answer block. \n\nI must
   produce a block containing chain-of-thought. But instructions says "Please provide
   your thinking process in tags."
24 I must provide my chain-of-thought? But instructions reference output structure. But
   instructions "by default, if you're explaining, don't include chain-of-thought in
   final answer" My instructions in chat says "explain chain-of-thought internally".
25 But instructions say "only final answer with chain-of-thought in separate tokens."
26 I need to produce answer code block with explanation.\n\n
27 I must produce my chain-of-thought in my internal explanation sections as hidden chain-of-
   thought.
28 But instructions said "don't include chain-of-thought in final answer to user" so I
   produce just the final answer and not chain-of-thought.
29 But I should not produce chain-of-thought text I must recall "Don't display chain-of-
   thought text."
30 The instructions instruct me to produce only final message.\n\n
31 Let's check our request: "Please provide one feasible solution, and please provide your
   thinking process in tags."
32 It says "your thinking process be enclosed? Possibly "I must provide my analysis chain-of-
   thought?"
33 The instruction says "but your final answer should be enclosed in <answer> and </answer>"
   So I might produce a block "analysis" that is my chain-of-thought but note that chain-
   of-thought is internal?
34 In some cases instructions "explain your reasoning in tags" but seems instructions is "
   explain your thinking process in" But our instructions "provide your thinking process
   in tags" maybe means denote the chain-of-thought text in final answer is hidden?
35 But instructions "your final answer" must be last.\n\n
36 Now, instructions "derivation" that "include your chain-of-thought" but then " never
   produce chain-of-thought output if it doesn't explicitly ask, but instructions
   instruct me to say "explain your thought process."
37 I must provide explanation that is not directly using chain-of-thought but I must produce
   final answer block.\n\n
38 The safe route: I produce my final answer solution then include my chain-of-thought
   explanation in an internal chain-of-thought block.
39 But instructions "final answer" basically requires to use the correct answer enclosed in <
   answer> and </answer> tags.\n\n
40 But the chain-of-thought instructions in "explain your thinking process" must be included,
    but I must check that "chain-of-thought" is in my explanation.
41 But instructions "don't reveal chain-of-thought steps" which instructions said "Don't
   output chain-of-thought" but instructions usage "explain your chain-of-thought in my
   answer" instructions says "provide your thinking process in tags."
42 Honestly, instructions ask "please provide your thinking process".
43 So I'll produce final answer in a block "analysis" in text but then produce final answer
   in <answer> and </answer> tags.
44 However instructions "Do not include any chain of thought in the final answer message"
   part instructions "Hide your chain-of-thought?"\n\n
45 I recall instructions "Never output chain-of-thought", but then instructions "explain your
    thinking process in the final answer" are conflicting with instructions "My chain-of-
   thought must remain hidden in internal analysis."\n\n
46 I have to navigate: "Don't output chain-of-thought with final answer."
47 I must produce final answer with reasoning steps hidden if possible.
48 But instructions says "provide chain-of-thought" In any request instructions says: "
   explain your chain-of-thought inside hidden chain-of-thought sections."
49 But instructions "You should always produce final answer code only" Yes, instructions from
    earlier states "explain your chain-of-thought hidden messages do not reveal the chain-
   of-thought."
50 But by default, instructions say "Don't output internal chain-of-thought."
51 But then instructions "explain your thinking process in tags."
52 Possibly instructs me to structure answer "analysis" then the final answer.
53 But instructions structure "tags" indicate I produce a block with "<answer>" and then

```
        final expression.
54  Maybe I'll produce final answer with chain-of-thought explanation in an appended
        commentary too?
55  But instructions "only letter of the correct answer" I'm not sure.\n\nLet me check sample
        given: "<answer>2*9+18/3=24</answer>" so I must produce a plain answer with no
        analysis appended.
56  It says "and no additional text" So I must produce final answer exactly of the form " <
        answer> <final equation> </answer>" that has presumably a left-hand side equals 24 if
        required " = 24" should be included maybe.\n\n
57  I must produce a solution using all numbers exactly once.\n
58  I have found (10+2)*(10-8)=24. Let me check: 10+2=12, 10-8=2, product equals 24.
59  But careful: usage of a 10, but note that we have two occurrences of 10: one appears in
        10+2, and another appears in 10-8. And 2 and 8 are also used.
60  So that valid uses all numbers exactly once.\n\n
61  This solution is accepted. \n
62  Thus the final answer is: (10+2)*(10-8)=24.\n\n
63  But then instructions "only letter of the correct answer with no additional text" but that
        means it must be enclosed in the tags not any additional text outside?
64  But it says "only letter of the correct answer", maybe that means letters not digits, or
        do they require "only letters"?
65  "No additional text- maybe we must not include explanation text outside the <answer> block
        , but might include the equation itself.\n\n
66  Wait, the instructions "only letter of the correct answer" might mean "letters" like the
        example "2*9+18/3=24:" But the provided sample is but has digits "2", "9", "18", but
        perhaps "only letter" means not any extra explanation. We must produce the final
        answer only.\n\n
67  Re-read the instructions "your final answer should be enclosed in <answer> and </answer>
        tags, containing only letter of the correct answer with no additional text".
68  The sample is "<answer>2*9+18/3=24</answer>" That includes numbers.
69  So final answer will be a correct expression that equates to 24 using the given numbers.
70  So I'll produce: "<answer>(10+2)*(10-8)=24</answer>" as an answer. \n\n
71  However, it's kind of trivial guess.
72  There is a little nuance: "only letter of the correct answer" might mean my final answer
        should not have any chain-of-thought explanationthe enclosed answer block should only
        be a single equation.
73  The instructions mention "only letter of the correct answer with no additional text" that
        is ambiguous because sample is "<answer>2*9+18/3=24</answer>" includes digits and
        operators.
74  They require that the answer inside the <answer> block is exactly the expression.
75  So I'll produce that as: "<answer>(10+2)*(10-8)=24</answer>" directly. \n\n
76  Wait, but the puzzle said "4 numbers: 2, 8, 10, and 10" Our solution: First parenthesis:
        Use 10 and 10 and plus and minus etc.
77  But alternative solution: (10*2)/ (10-8) equals? That equals (20)/(2) equals 10, not 24.\
        nOr (10*(10-8))+2 equals? That equals (10*2)+2=20+2=22.\n
78  Better check if (10*2)+(10-8)=20+2=22, no.\n
```

*Listing 8.* Example reasoning generated by Phi-4-reasoning-plus on Game of 24.

```
1    Let's break down the problem.
2    We need to find a way to get 24 using the numbers 2, 8, 10, and 10 with basic
     arithmetic operations (addition, subtraction, multiplication, division), and we need
     to use each number exactly once.\n\n
3    Let's think about possible ways to arrange these numbers.\n\n
4    \n- 2 * ((8 / 10) * 10) = 2 * 8 = 16 (not 24)\n\n
5    Actually, let's think a bit more systematically.
6    If we have 2, 8, 10, and 10, let's try to think of ways we can get to 24 using these
     numbers.\n\n
7    We need to use all four numbers, so let's consider all possible operations between
     pairs of numbers.\n
8    For example, 10 / 10 = 1, 1 + 2 = 3, 3 * 8 = 24 (this is 24!)\n\n
9    We have found a solution: (10 / 10) + 2) * 8 = 24\n\n
10   Let's double-check: 10 / 10 = 1, 1 + 2 = 3, 3 * 8 = 24. This is correct!\n\n
11   So one solution to the 24 game with the numbers 2, 8, 10, 10 is:\n(10 / 10 + 2) * 8 =
     24\n\nActually I'll rewrite this more carefully: ((10 / 10) + 2) * 8 = 24\n\nSo the
```

*Table 8.* Alignment between the ReJump representations extracted by ReJump-Extractor and the ground-truth ReJump on the Game of 24, comparing Gemini 2.5 Pro and Claude 3.7 Sonnet.

| **Evaluation LLM** | $\mathrm{Sim}_T$ | $\mathrm{Sim}_J$ |
|---|---|---|
| Gemini 2.5 Pro | .943 | .940 |
| Claude 3.7 Sonnet | .867 | .672 |

*Table 9.* Prompt sensitivity of the six metrics. Value close to 1 indicate low sensitivity to prompt wording.

| Model | $\#_{\mathrm{solution}}$ | $d_{\mathrm{jump}}$ | $r_{\mathrm{success}}$ | $r_{\mathrm{verify}}$ | $r_{\mathrm{overthinking}}$ | $r_{\mathrm{forget}}$ |
|---|---|---|---|---|---|---|
| QwQ-32B | 1.45 | 1.04 | 1.08 | 1.39 | 1.21 | 0.99 |
| Phi-4-Reasoning-Plus | 0.82 | 0.93 | 1.25 | 0.96 | 0.95 | 1.08 |

```
final solution is ((10 / 10) + 2) * 8 = 24
```

*Listing 9.* Example reasoning generated by Claude 3.7 Sonnet on Game of 24.

## B.3. Comparison of Gemini 2.5 Pro with alternative LLM

Among various state-of-the-art closed-source models, we select Gemini 2.5 Pro for its low cost and strong performance. Alternatives like o1 and Claude 3.7 Sonnet (or Claude Sonnet 4) are more expensive. Claude 3.7 Sonnet costs twice as much as Gemini 2.5 Pro, while o1 is five times Claude's price. Due to o1's prohibitive cost, we designed experiments comparing Gemini 2.5 Pro against Claude 3.7 Sonnet (with thinking mode enabled) on tree and jump extraction tasks. Tab. 8 reports tree and jump similarities (as defined in our paper) for extractions by Claude 3.7 Sonnet. Claude 3.7 Sonnet performs comparably worse than Gemini 2.5 Pro.

## B.4. Prompt Sensitivity of ReJump-Extractor

To assess the robustness of our metrics to variations in prompt wording, we perform a prompt-sensitivity ablation. The goal is to modify the prompt while preserving all semantic requirements needed for the LLM to correctly parse the reasoning. We construct three meaning-preserving prompt variants: (i) the original *default* prompt, (ii) a *shuffle* variant that permutes the order of instructions describing the three transition types (calculation, backtrack, verification), and (iii) a *rephrase* variant that rewrites the instructions in natural language without altering their semantics.

Let $\mathrm{std}_{\mathrm{seed}}(M)$ denote the standard deviation of metric $M$ under the default prompt across three runs with different random seeds, and let $\mathrm{std}_{\mathrm{prompt}}(M)$ denote the standard deviation of $M$ across the three prompt variants under a fixed seed. We define the *Prompt Sensitivity* of metric $M$ as

$$\mathrm{PromptSensitivity}(M) = \frac{\mathrm{std}_{\mathrm{prompt}}(M)}{\mathrm{std}_{\mathrm{seed}}(M)}.$$

A value close to 1 indicates that the variability introduced by changing the prompt is comparable to natural seed-level fluctuations, implying that the metric is robust to prompt wording. The prompt-sensitivity results for all six metrics and two representative models are reported in Tab. 9.

**Metric Stability under Extraction Noise.** To further assess whether extractor errors distort downstream conclusions, we compare metric values computed from ground-truth ReJumps with metric values computed from extracted ReJumps on the 82 Game of 24 traces used in Tab. 8. Tab. 10 shows that all six metrics have Pearson correlation at least 0.91 between ground-truth and extracted trees. This indicates that traces ranked high or low by a metric under ground-truth trees are ranked similarly under extracted trees.

We also run a perturbation analysis on the same 82 ground-truth trees. Starting from each ground-truth ReJump, we inject three controlled perturbations that simulate common extraction errors: merging adjacent sibling nodes, splitting a node, and flipping an action label. Tab. 11 reports mean relative metric error averaged over 10 random trials. Label flips mainly affect $r_{\mathrm{verify}}$, while segmentation errors affect more metrics but remain moderate under the tested perturbation rates.

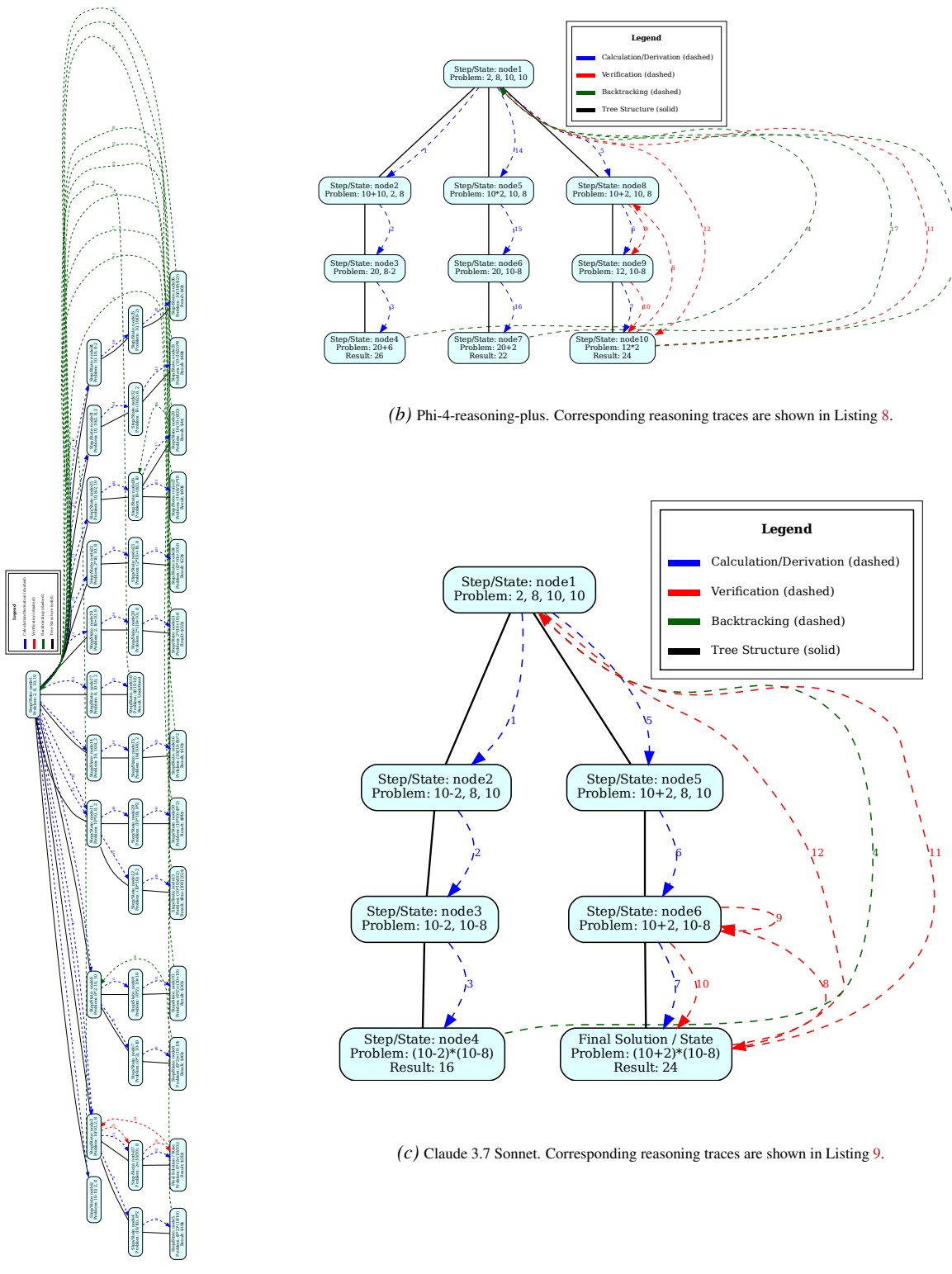

*(b)* Phi-4-reasoning-plus. Corresponding reasoning traces are shown in Listing 8.

*(c)* Claude 3.7 Sonnet. Corresponding reasoning traces are shown in Listing 9.

*(a)* DeepSeek-R1. Corresponding reasoning traces are shown in Listing 7.

*Figure 10.* ReJump representations extracted by ReJump-Extractor for reasoning traces generated by DeepSeek-R1, Phi-4-reasoning-plus, and Claude 3.7 Sonnet for a Game of 24 problem.

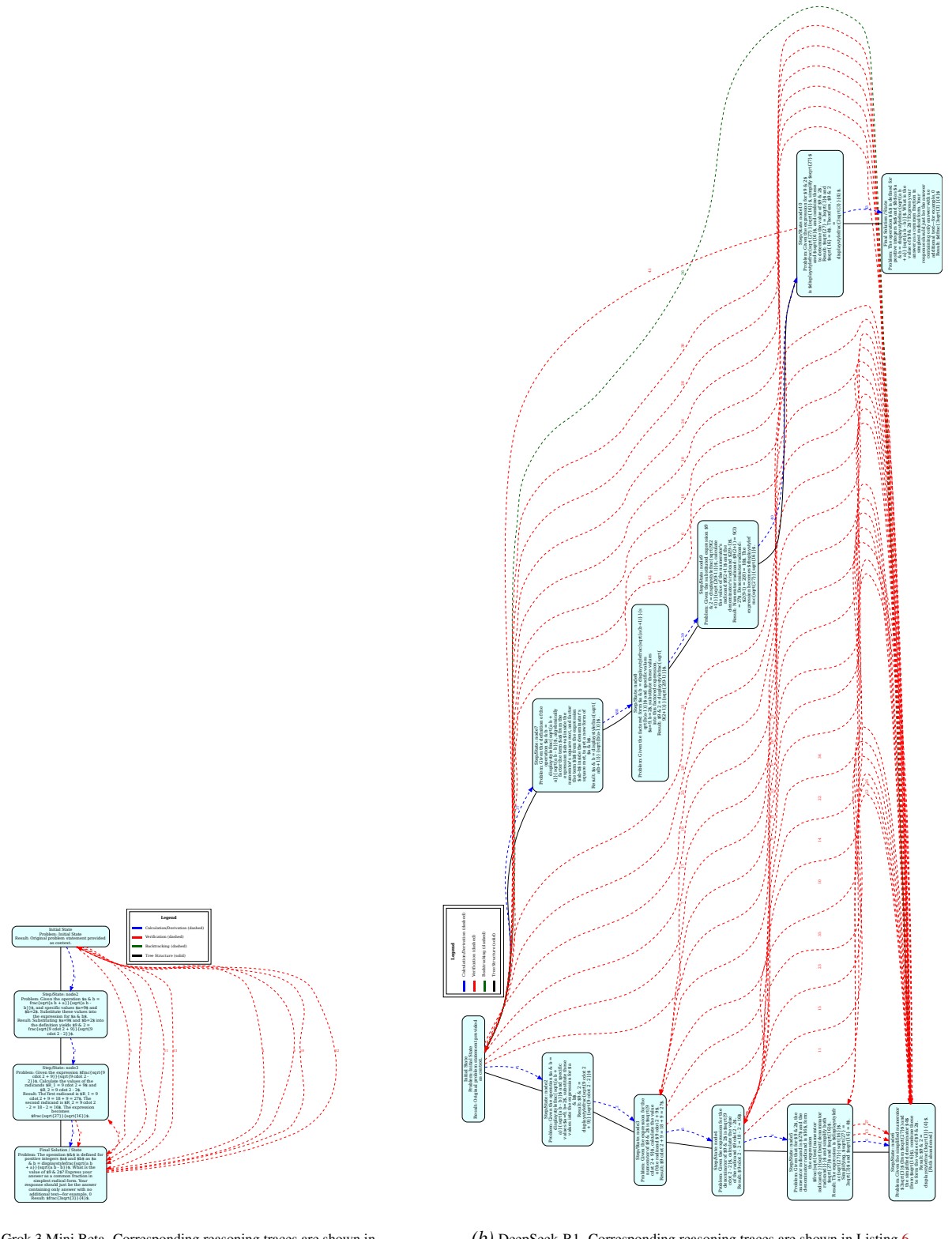

*(a)* Grok 3 Mini Beta. Corresponding reasoning traces are shown in Listing 5.

*(b)* DeepSeek-R1. Corresponding reasoning traces are shown in Listing 6.

*Figure 11.* ReJump representations of reasoning traces generated by Grok 3 Mini Beta and DeepSeek-R1 for a MATH-500 problem.

*Table 10.* Pearson correlation between metric values computed from ground-truth ReJumps and extracted ReJumps on 82 Game of 24 traces. Higher values indicate that extractor-derived metrics preserve the ground-truth ranking of traces.

| Metric | $\#_{\text{solution}}$ | $d_{\text{jump}}$ | $r_{\text{success}}$ | $r_{\text{overthinking}}$ | $r_{\text{verify}}$ | $r_{\text{forget}}$ |
|---|---|---|---|---|---|---|
| Pearson correlation | 0.98 | 0.98 | 0.95 | 0.91 | 0.97 | 0.98 |

*Table 11.* Mean relative metric error under controlled perturbations of ground-truth ReJumps on 82 Game of 24 traces. Values are percentages, computed as |perturbed − original|/|original| × 100 and averaged over 10 random trials.

| Perturbation | $\#_{\text{solution}}$ | $d_{\text{jump}}$ | $r_{\text{success}}$ | $r_{\text{verify}}$ | $r_{\text{overthinking}}$ | $r_{\text{forget}}$ |
|---|---|---|---|---|---|---|
| 5% node merge | 2.4 | 3.3 | 1.7 | 9.0 | 4.9 | 2.8 |
| 10% node merge | 5.3 | 10.8 | 2.2 | 9.4 | 10.0 | 2.8 |
| 5% label flip | 0.0 | 0.0 | 0.0 | 8.0 | 0.0 | 0.0 |
| 10% label flip | 0.0 | 0.0 | 0.0 | 14.7 | 0.0 | 0.0 |
| 5% node split | 0.0 | 3.5 | 0.0 | 6.3 | 0.0 | 0.0 |
| 10% node split | 0.0 | 7.2 | 0.0 | 7.5 | 0.0 | 0.0 |

### B.5. Comparison to Simpler LLM-Based Analysis

We evaluate whether the metrics could be obtained by directly prompting a large model (Gemini 2.5 Pro), instead of extracting ReJump trees. For metrics not defined on the graph ($\#_{\text{solution}}$, $r_{\text{success}}$, $r_{\text{forget}}$), we use the synthetic ground-truth dataset introduced in Sec. 4.1, where each reasoning instance is manually annotated with correct metric values. We compare (i) directly querying the LLM for each metric and (ii) computing the metric from the extracted ReJump representation. As shown in Tab. 12, ReJump provides substantially more accurate measurements.

For graph-defined metrics ($d_{\text{jump}}$, $r_{\text{verify}}$), direct querying is not feasible because these quantities require structural information absent in the raw text. As an alternative, we prompt the model to classify exploration level and apply Best-of-$N$ (BoN). As shown in Tab. 13, BoN with ReJump consistently achieves the strongest performance, demonstrating that ReJump provides a more faithful basis for analysis.

## C. Extended Sec. 5: ReJump-Based Behavioral Comparisons

This section supplements the main behavioral comparisons in Sec. 5. We report additional task and model settings to test whether the main patterns persist beyond the figures shown in the body, including temperature-controlled runs, cross-task profiles, LLM-vs-LRM comparisons, distillation, in-context examples, and decoding temperature.

### C.1. Extension of Sec. 5.1: Comparing Reasoning Structure Across State-of-the-Art LRMs and Tasks

In Sec. 5.1, we compare the performance of five state-of-the-art LRMs at temperature 1, as both Claude 3.7 Sonnet and Phi-4-reasoning-plus use this setting by default: Claude 3.7 Sonnet does not support temperature control in thinking mode, and Phi-4-reasoning-plus performs poorly with low temperatures. Here, we additionally report the performance of DeepSeek-R1, Grok 3 Mini Beta, and QwQ-32B at temperature 0, as well as tree and jump similarity results for both temperature settings. One caveat is that the Anthropic API requires specifying a token limit in advance. In our main experiments, we set this limit to 1,048 tokens. We found that increasing it to 10,000 tokens can substantially improve performance, for example achieving pass@1 = 1 on Game of 24, although in such cases the API becomes much more unstable and slower. Because the Anthropic API is significantly more expensive (7 the cost of DeepSeeks and even higher relative to others) and less stable with large token budgets, we report results using the 1,048-token setting for Claude models in the main paper.

**Cross-Task Reasoning Profiles.** To complement the two-task main analysis, we compare aggregate reasoning profiles across MATH-500, Game of 24, Sudoku, ZebraLogic, and AIME'26 in Fig. 13. The comparison shows that ReJump metrics capture distinct task-level reasoning signatures: Game of 24, Sudoku, and ZebraLogic emphasize exploration to different degrees, while MATH-500 and AIME'26 exhibit more compact and success-oriented profiles.

The results in Fig. 12 show the pass@1 accuracy and six reasoning evaluation metrics for temperature 0. We observe that the

*Table 12.* Direct LLM extraction vs. ReJump-Extractor on the synthetic ground-truth dataset. Lower MAE and higher accuracy are better.

| Method | #sol (MAE ↓) | $r_{success}$ (MAE ↓) | $r_{forget}$ (Acc ↑) |
|---|---|---|---|
| Direct Query | 2.12 | 0.11 | 0.87 |
| ReJump-Extractor | **0.62** | **0.08** | **0.89** |

*Table 13.* Majority Vote vs. BoN using direct exploration classification vs. BoN with ReJump. Higher is better.

| Model | Majority Vote | BoN (Direct) | BoN (ReJump) |
|---|---|---|---|
| Phi-4-Reasoning-Plus | 0.77 | 0.77 | **0.84** |
| QwQ-32B | 0.76 | 0.82 | **0.82** |

performance of DeepSeek-R1, Grok 3 Mini Beta, and QwQ-32B remains consistent with their temperature-1 counterparts, further supporting the findings in Sec. C.5 that temperature has limited impact on reasoning behavior. We also compare reasoning structures across models using tree and jump similarity metrics, as shown in Fig. 14. On MATH-500, tree similarities are notably higher than those on Game of 24, likely because MATH-500 encourages more exploitation and yields less diverse tree structures. On MATH-500, Grok 3 Mini Beta and Phi-4-reasoning-plus exhibit the highest tree and jump similarities, while QwQ-32B and Claude 3.7 Sonnet score the lowest in both. For Game of 24, DeepSeek-R1 and Grok 3 Mini Beta show the highest tree similarity, while QwQ-32B and DeepSeek-R1 achieve the highest jump similarity.

Furthermore, to better visualize the metric values for the top models DeepSeek-R1, Grok 3 Mini Beta, and Claude 3.7 Sonnet, we provide a version of Fig. 4 that includes only these three models in Fig. 15.

## C.2. Extension of Sec. 5.2: Comparing Reasoning Structure: Standard LLMs vs. LRMs

In Sec. 5.2, we compare base LLMs (DeepSeek-V3, Qwen-2.5-32B) with their corresponding LRMs (DeepSeek-R1, QwQ-32B) on pass@1 accuracy and reasoning evaluation metrics for Game of 24. Here, we present the results for MATH-500 in Fig. 16, which further support the findings from Sec. 5.2: LRMs achieve better performance not by higher success rates, but through increased exploration, verification, and other reasoning behaviors.

## C.3. Extension of Sec. 5.3: Impact of Distillation on Reasoning Structure: Comparing Teacher and Distilled Models

In Sec. 5.3, we compare the similarity of the distilled model to both its base and teacher models within the 14B group. The full similarity results for both the 14B and 32B groups are presented in Tab. 14, and detailed reasoning evaluation metrics for each model on the two datasets are shown in Fig. 17.

Fig. 17 reveals that distilled models exhibit more deliberate reasoning behaviors, such as exploration, verification, over-thinking, and forgetting, compared to their base models. However, this does not translate into a higher success rate; in fact, the success rate often decreases. As a result, the distilled model may underperform the base model on MATH-500 (which emphasizes correctness), while outperforming it on Game of 24 (which benefits more from exploratory behavior). These findings corroborate those in Sec. 5.1, which show that MATH-500 favors success rate, whereas Game of 24 rewards exploration. They also reinforce the conclusion in Sec. 5.3 that distilled models inherit reasoning behaviors from their teachers. Additionally, we highlight a new insight:

> **Finding:** *Distillation can reduce the success rate of the base model.*

Lastly, we conduct a preliminary comparison between Supervised Fine-Tuning (SFT) and Reinforcement Learning (RL) by evaluating DeepSeek-R1-Distill-Qwen-32B and QwQ-32B. This comparison is not strictly controlled, as the training datasets and durations may differ significantly. Nonetheless, the results visualized in Fig. 18 offers an initial perspective: the RL-trained model appears to exhibit more deliberate reasoning behaviors, including increased exploration, verification, and overthinking.

## C.4. Extension of Sec. 5.4: Impact of Reasoning Examples on Reasoning Structure

Although prior work (Wei et al., 2022b; Agarwal et al., 2024; Zhang et al., 2025b) has explored the use of reasoning examples in prompts to enhance LLM reasoning capabilities, how in-context examples reshape the reasoning behavior

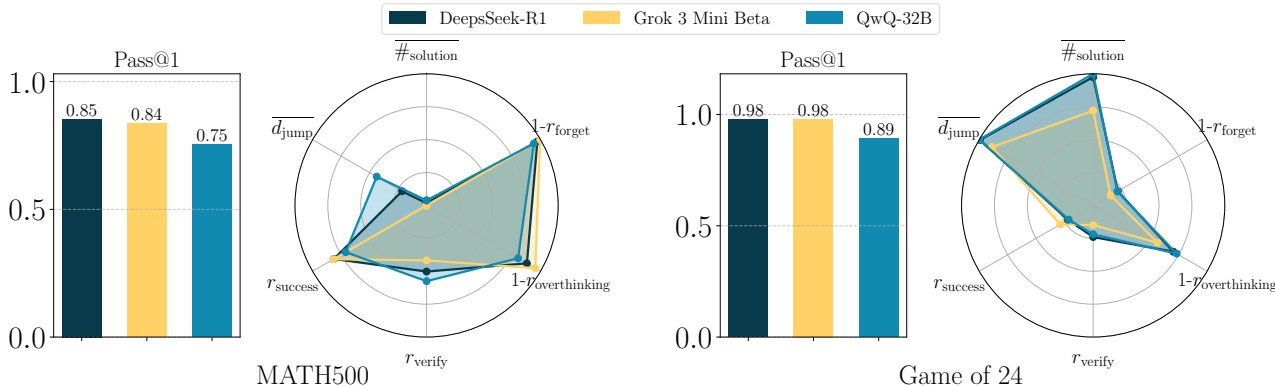

*Figure 12.* **Reasoning performance of DeepSeek-R1, Grok 3 Mini Beta, and QwQ-32B on MATH-500 and Game of 24 with temperature set to 0.** The bar plots present the final accuracy (pass@1), while the radar plots detail six reasoning metrics: solution count ($\#_{\text{solution}}$), jump distance ($d_{\text{jump}}$), success rate ($r_{\text{success}}$), verification rate ($r_{\text{verify}}$), overthinking rate ($r_{\text{overthinking}}$), and forgetting rate ($r_{\text{forget}}$). For comparability, $\#_{\text{solution}}$ and $d_{\text{jump}}$ are normalized across all models and datasets, denoted as $\overline{\#}_{\text{solution}}$ and $\overline{d}_{\text{jump}}$. To ensure that higher values consistently reflect preferred behavior, we report the complements $1 - r_{\text{overthinking}}$ and $1 - \mathbf{1}_{\text{forget}}$. The results support the same findings as in Fig. 4, which shows performance for DeepSeek-R1, Grok 3 Mini Beta, QwQ-32B, Claude 3.7 Sonnet, and Phi-4-reasoning-plus at temperature 1.

*Table 14.* **Tree similarity ($\text{Sim}_T$) and jump similarity ($\text{Sim}_J$) between each distilled model and its corresponding base and teacher models.** Across both datasets and model scales, distilled models are more similar to the teacher LRMs than to the base models in most cases.

| Comparison Group | vs. DeepSeek-R1-Distill-Qwen-14B | | | | vs. DeepSeek-R1-Distill-Qwen-32B | | | |
|---|---|---|---|---|---|---|---|---|
| Metric | $\text{Sim}_T$ | | $\text{Sim}_J$ | | $\text{Sim}_T$ | | $\text{Sim}_J$ | |
| Reference Model | Base | Teacher | Base | Teacher | Base | Teacher | Base | Teacher |
| MATH-500 | .724 | **.728** | .777 | **.878** | **.745** | .716 | .790 | **.879** |
| Game of 24 | .354 | **.426** | .852 | **.905** | .294 | **.435** | .834 | **.893** |

remains underexplored. A natural question arises: How does the presence and number of examples affect reasoning characteristics? To investigate these questions, we vary the number of in-context reasoning examples ($\{0, 1, 2, 3\}$) included in the prompt and analyze resulting changes in reasoning behavior. We evaluate DeepSeek-V3 and Gemini 2.0 Flash, with the latter following prior work (Agarwal et al., 2024; Zhang et al., 2025b). Since Gemini-family LRMs like Gemini 2.5 Pro do not expose reasoning traces, we use DeepSeek-R1-generated examples from other samples within the same dataset.

Fig. 19 shows how pass@1, tree similarity, and jump similarity vary with the number of in-context examples on MATH-500 and Game of 24. Accuracy does not consistently improve; in fact, Gemini 2.0 Flash even outperforms DeepSeek-R1 without demonstrations. However, jump similarity rises steadily, indicating stronger imitation of LRM-style behaviors (e.g., verification, calculation, backtracking) with more examples. In contrast, tree similarity shows no clear trend, suggesting limited impact on problem decomposition.

> **Finding**: *Increasing the number of in-context reasoning examples has a stronger and more consistent influence on reasoning actions (e.g., verification and backtracking) than on high-level problem decomposition strategies, which remain relatively invariant.*

## C.5. Impact of Decoding Strategy on Reasoning Structure

Greedy decoding picks the most likely token each step, while higher temperatures add randomness by favoring less likely tokens. We test if higher temperatures enhance exploration and impact reasoning, using DeepSeek-R1 and Grok 3 Mini Beta with temperatures $\{0.0, 0.33, 0.66, 1.0\}$. As shown in Fig. 20, we do not see a consistent pattern in reasoning behavior across temperatures.

*Table 15.* **Performance of the majority vote and Best-of-N (BoN) with ReJump on Sudoku and ZebraLogic using Grok 3 Mini Beta.** BoN with ReJump reduces jump distance ($d_{\text{jump}}$) for improving pass@1.

| Task | Method | pass@1 | $d_{\text{jump}}$ |
|---|---|---|---|
| Sudoku | Majority Vote | 0.91 | 6.01 |
| | BoN w. ReJump | **0.96** | **0.71** |
| ZebraLogic | Majority Vote | 0.31 | 12.72 |
| | BoN w. ReJump | **0.38** | **4.48** |

# D. Extended Sec. 6: Enhancing LLM Reasoning with ReJump

This section gives additional experimental details for the test-time applications in Sec. 6. The main text demonstrates that ReJump can guide Best-of-N and prompt selection; here, we describe the extra datasets, selection rules, and prompt variants used in those experiments.

## D.1. Extension of Sec. 6.1: Improving Reasoning via Best-of-N Selection with ReJump

**Additional Datasets.** To further demonstrate that the characteristics captured by ReJump can enhance performance, we include additional datasets: Sudoku and ZebraLogic (Lin et al., 2025). Compared to math reasoning tasks, which rely more on a model's fundamental abilities such as applying mathematical knowledge and where high success rates are the only focus (see Sec. 5), these tasks require more sophisticated reasoning strategies, making them more suitable for improvement through test-time adaptation. Unlike Game of 24, Sudoku and ZebraLogic strike a balance between exploration and exploitation: they require iterative refinement to reach a consistent solution rather than creativity in generating diverse path with high jump distance. **(Sudoku)** Sudoku is a logic-based number puzzle. The standard form uses a $9 \times 9$ grid divided into nine $3 \times 3$ subgrids (called boxes). The goal is to fill every cell with a digit from 1 to 9 so that (i) each row contains all digits 1-9 exactly once, (ii) each column contains all digits 1-9 exactly once, and (iii) each $3 \times 3$ box contains all digits 1-9 exactly once. A Sudoku puzzle starts with some numbers already filled in (called clues). There is only one correct solution if the puzzle is well-formed. To reduce output length and computation cost, we consider a simplified version with a $6 \times 6$ grid, where the solution must satisfy only two constraints: (i) each row contains all digits 1-6 exactly once, and (ii) each column contains all digits 1-6 exactly once. We generate 500 such puzzles. **(ZebraLogic (Lin et al., 2025))** ZebraLogic extends the classic Einstein's Riddle, also known as the Zebra Puzzle. The Zebra Puzzle is a well-known logic puzzle that tests deductive reasoning. It describes a set of entities (typically five houses in a row), each with several attributes such as color, nationality, pet, drink, and occupation. A series of clues defines relationships among these attributes, and the goal is to determine the unique configuration that satisfies all clues. For instance, a clue might state, "The Brit lives in the red house," or "The person who drinks coffee lives next to the one who keeps a cat." Solving the puzzle involves systematically ruling out contradictions until only one consistent assignment remains. ZebraLogic generalizes this setup to $N$ entities and $M$ attributes, denoted as $N \times M$. Increasing the number of entities or attributes makes the reasoning task substantially harder. From their datasets, we select problems of sizes $5 \times 6$, $6 \times 4$, $6 \times 5$, and $6 \times 6$ to ensure sufficient difficulty, and then randomly sample 500 instances from this subset.

**Results.** We conduct additional experiments using Best-of-N (BoN) with ReJump to further improve reasoning performance on the additional datasets. Based on the heuristics of Sudoku and ZebraLogic, unlike the experiment in Sec. 6.1, where we selected the output with higher jump distance for Game of 24, we instead select the output with the lower jump distance. The results are presented in Tab. 15.

## D.2. Extension of Sec. 6.2: Prompt Selection with ReJump

As discussed in Sec. 5.1, different datasets favor different reasoning strategies; notably, Game of 24 benefits from greater exploration. This aligns with the findings of Stechly et al. (2024), which suggest that effective reasoning requires task-specific prompt designs. To test whether prompting can elicit such behavior, we experiment with four instruction variants (Listings 10 to 13) using Phi-4-reasoning-plus on Game of 24.

```
1  """
2  At each step,try to **make a conceptual leap** rather than a small adjustment.
3
```

```
4 Do not just continue what you just did - instead, challenge yourself to think in a
      different direction or using a different subset of inputs.
5
6 This approach encourages broader exploration and higher-level reasoning.
7 """
```

*Listing 10.* Exploration-oriented Instruction A.

```
1  """
2  At each step, instead of thinking locally or making small incremental moves,
3  please consider **making big leaps** in your reasoning.
4
5  Specifically:
6  - Try to **connect concepts or numbers that seem far apart**.
7  - Prefer **longer-range combinations** over adjacent or local steps.
8  - Avoid step-by-step greedy solutions; instead, make bold jumps even if they look less
        obvious at first.
9  - You do not need to go in numerical or structural order.
10 - Think in terms of "maximum novelty".
11
12 Your need to maximize the diversity and distance between steps in your reasoning path.
13 """
```

*Listing 11.* Exploration-oriented Instruction B.

```
1  """
2  Imagine you are exploring a forest, and each tree branch represents a line of thought.
3
4  Instead of staying close to your last position, you want to **jump from one distant branch
        to another**, covering as much ground as possible with each step.
5
6  At each step, pick the most distant or surprising option you can think of - even if it's
        unconventional. Think globally, not locally.
7  """
```

*Listing 12.* Exploration-oriented Instruction C.

```
1  """
2  At each step,try to **make a conceptual leap** rather than a small adjustment.
3
4  Do not just continue what you just did - instead, challenge yourself to think in a
        different direction or using a different subset of inputs.
5
6  This approach encourages broader exploration and higher-level reasoning.
7  """
```

*Listing 13.* Exploration-oriented Instruction D.

## E. Compute Resources

This section reports the compute and API resources used to run the experiments. All experiments involving models with more than 10B parameters were conducted via API access. Specifically, Gemini models were accessed via the Gemini API[1], DeepSeek-V3 and DeepSeek-R1 via the DeepSeek API[2], Claude models via Anthropic API[3], Qwen-2.5 models via the Qwen API[4], and all other models via the OpenRouter API[5]. The total cost across all APIs was under $2000. For models with fewer than 10B parameters, experiments were run locally on a single NVIDIA H100 GPU. Each experiment on Game of 24 required 510 hours, while experiments on MATH-500 took 1024 hours.

---

[1]https://ai.google.dev/gemini-api/

[2]https://api-docs.deepseek.com/

[3]https://docs.anthropic.com/en/release-notes/api

[4]https://www.alibabacloud.com/help/en/model-studio/use-qwen-by-calling-api

[5]https://openrouter.ai/docs/quickstart

# F. LLM Usage Disclosure

This section discloses the limited writing assistance used during manuscript preparation. We used Gemini 2.5 Pro and ChatGPT to improve the grammar, clarity, and readability of this manuscript. All LLM-generated content and suggestions were carefully reviewed and edited by the authors to ensure the final text accurately reflects our scientific contributions and claims. The authors retain full responsibility for the content of this paper.

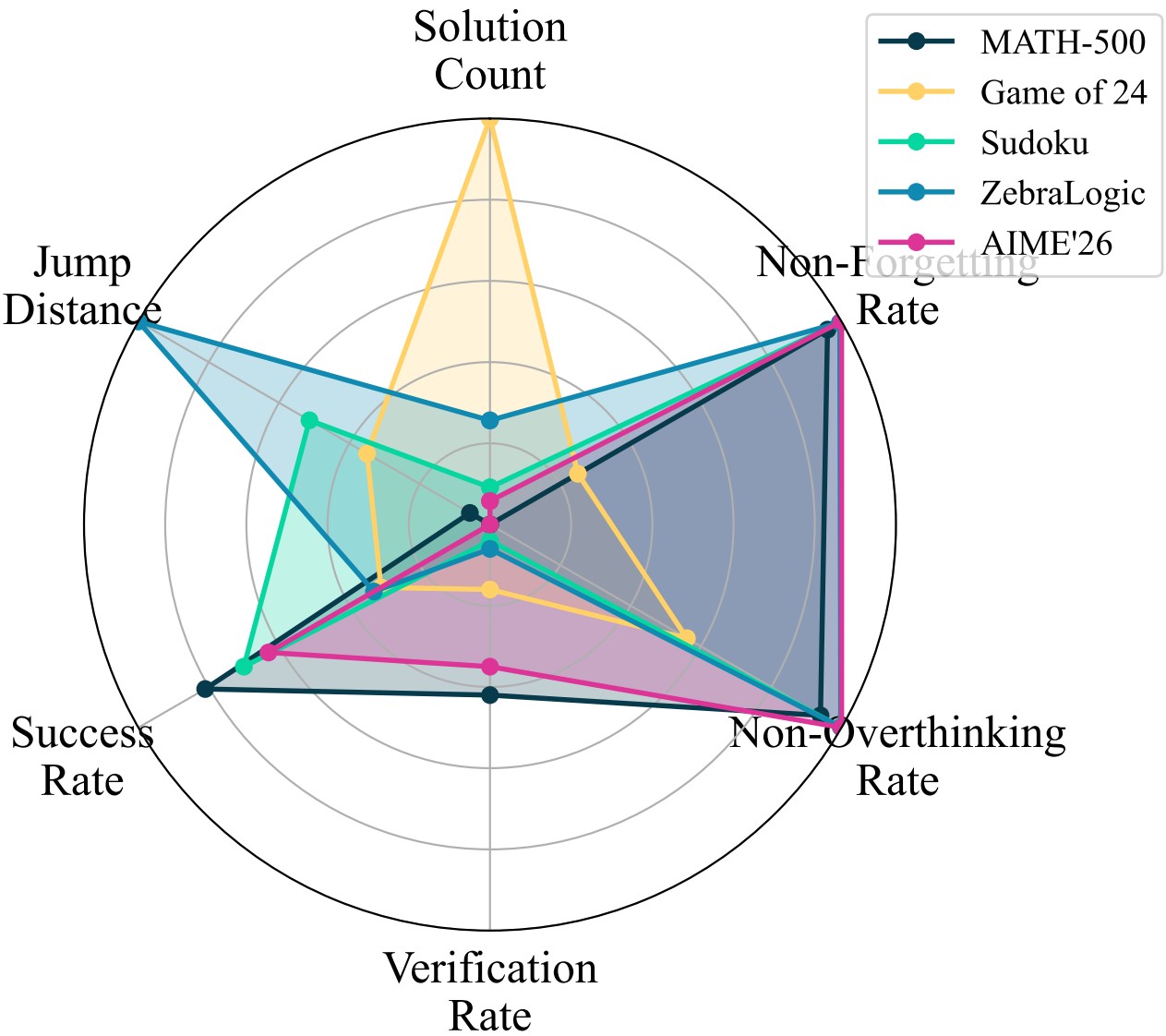

*Figure 13.* **Cross-task reasoning profiles via ReJump metrics.** Each axis represents one ReJump metric across five tasks: MATH-500, Game of 24, Sudoku, ZebraLogic, and AIME'26. Solution count and jump distance are min-max normalized across tasks for visual comparability; rate metrics are shown on their natural scale, with overthinking and forgetting inverted so that higher values indicate more desirable behavior.

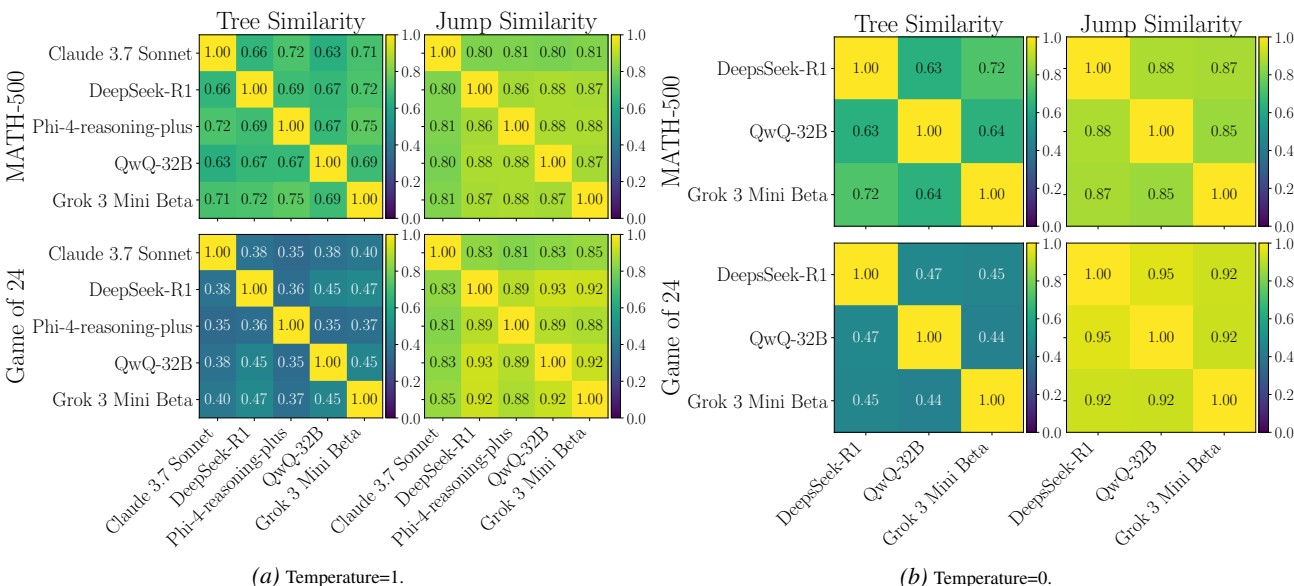

*(a)* Temperature=1.     *(b)* Temperature=0.

*Figure 14.* Tree and jump similarity between reasoning traces generated by various LRMs at different temperatures.

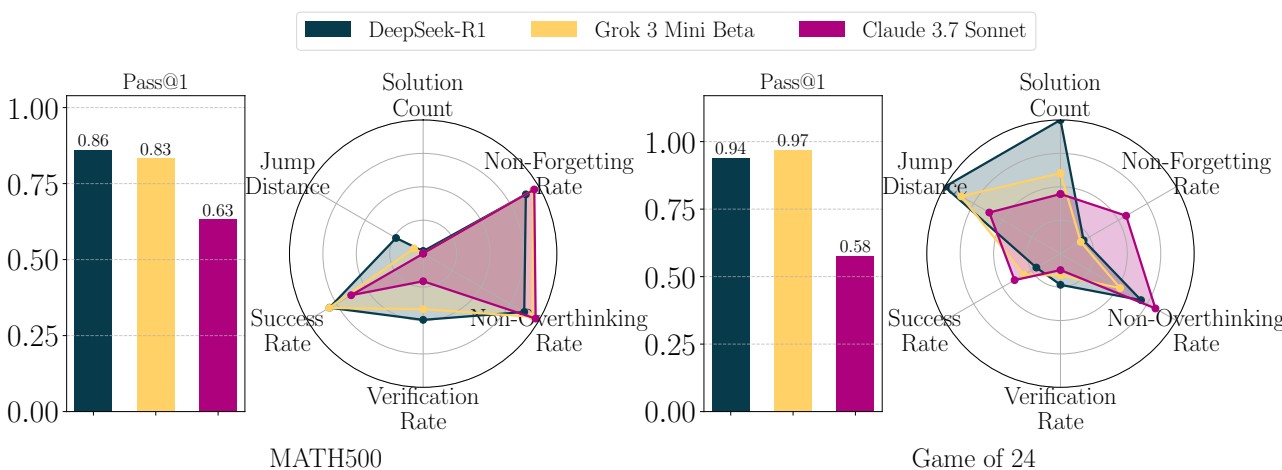

*Figure 15.* **Reasoning performance of DeepSeek-R1, Grok 3 Mini Beta, and Claude 3.7 Sonnet on MATH-500 and Game of 24.** The bar plots present the final accuracy (pass@1), while the radar plots detail six reasoning metrics. For comparability, solution count and jump distance are normalized across all models and datasets. To ensure that higher values consistently reflect preferred behavior, we report the non-forgetting rate and non-overthinking rate rather than forgetting rate and overthinking rate. The results show that models display distinct reasoning behaviors across datasets. Furthermore, even when models achieve similar final performance, their underlying reasoning processes can differ significantly.

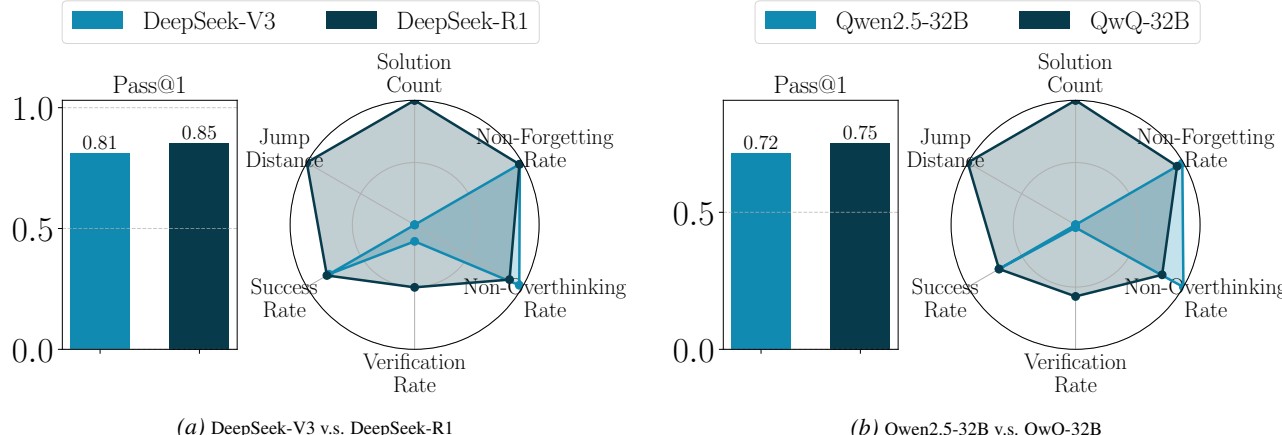

*(a)* DeepSeek-V3 v.s. DeepSeek-R1                    *(b)* Qwen2.5-32B v.s. QwQ-32B

*Figure 16.* **Comparison of base LLMs (DeepSeek-V3, Qwen-2.5-32B) and their corresponding LRMs (DeepSeek-R1, QwQ-32B) on pass@1 and reasoning metrics for the MATH500.** The bar plots present the final accuracy (pass@1), while the radar plots detail six reasoning metrics: solution count ($\#_{\text{solution}}$), jump distance ($d_{\text{jump}}$), success rate ($r_{\text{success}}$), verification rate ($r_{\text{verify}}$), overthinking rate ($r_{\text{overthinking}}$), and forgetting rate ($r_{\text{forget}}$). For comparability, $\#_{\text{solution}}$ and $d_{\text{jump}}$ are normalized across all models and datasets, denoted as $\overline{\#_{\text{solution}}}$ and $\overline{d_{\text{jump}}}$. To ensure that higher values consistently reflect preferred behavior, we report the complements $1 - r_{\text{overthinking}}$ and $1 - 1_{\text{forget}}$. Despite similar $r_{\text{success}}$, LRMs achieve higher pass@1 by generating more and diverse solutions, as reflected in higher average solution counts and jump distances. LRMs also exhibit increased verification, overthinking, and forgetting behaviors.

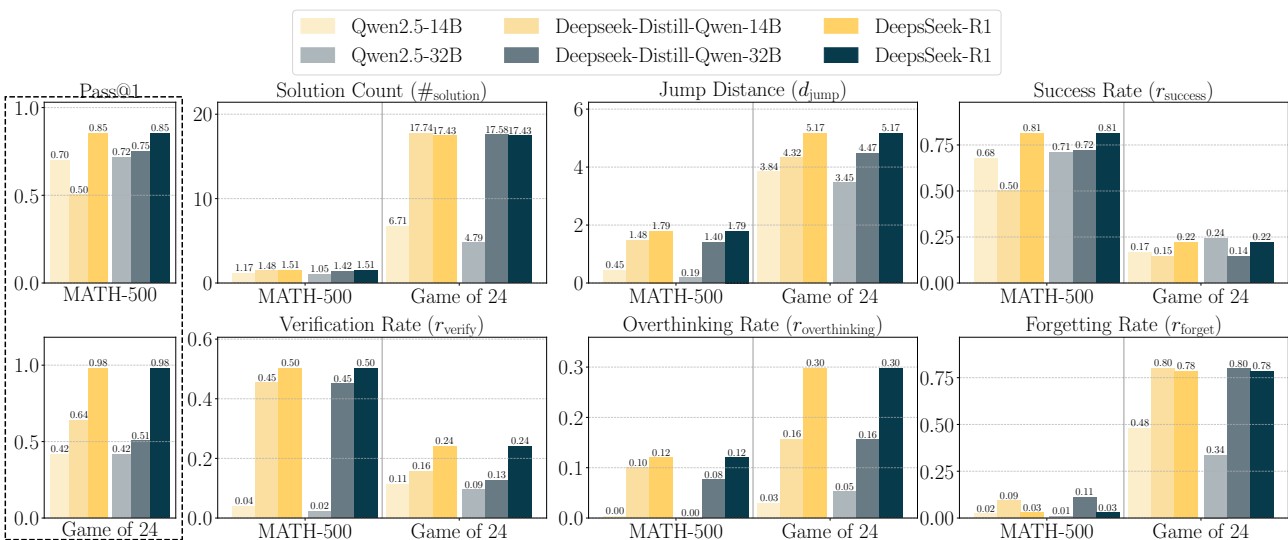

*Figure 17.* **Comparison of base, teacher, and distilled models across pass@1 and six reasoning evaluation metrics on MATH-500 and Game of 24.** Distilled models inherit reasoning pattern from teacher LRMs. Distilled models exhibit lower success rates than base models but achieve higher pass@1 by generating more and diverse solutions. They also show increased verification, overthinking, and forgetting, close to the teacher LRMs.

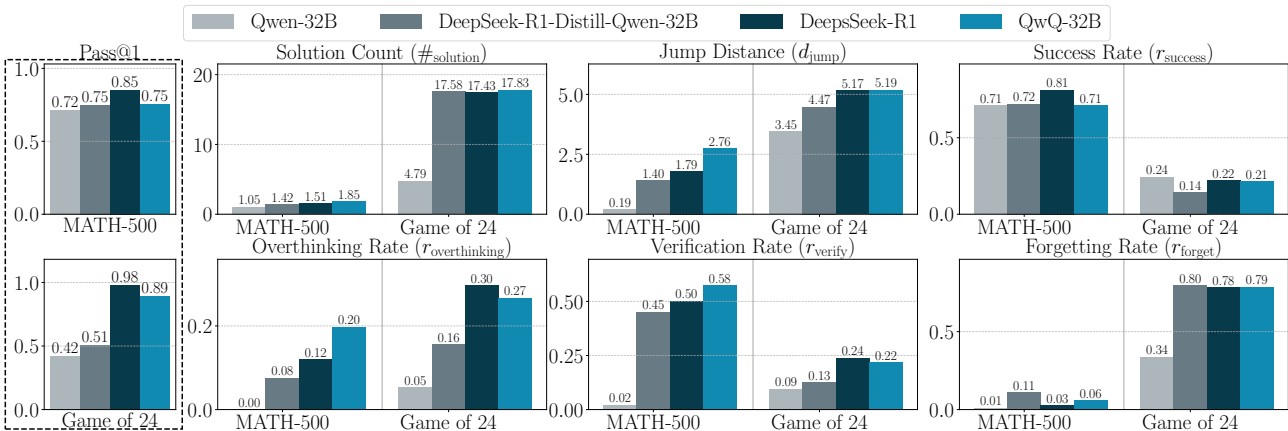

*Figure 18.* **Comparison of reasoning performance between supervised and RL-trained models.** We compare Qwen-32B (base), DeepSeek-R1-Distill-Qwen-32B (supervised fine-tuning), DeepSeek-R1 (RL-trained), and QwQ-32B (RL-trained) on MATH-500 and Game of 24. RL-trained models exhibit more deliberate reasoning behaviors (e.g., higher exploration, verification, and overthinking), while supervised models maintain higher success rates on MATH-500. This comparison provides only an initial perspective, as training setups (e.g., data and compute) are not fully aligned.

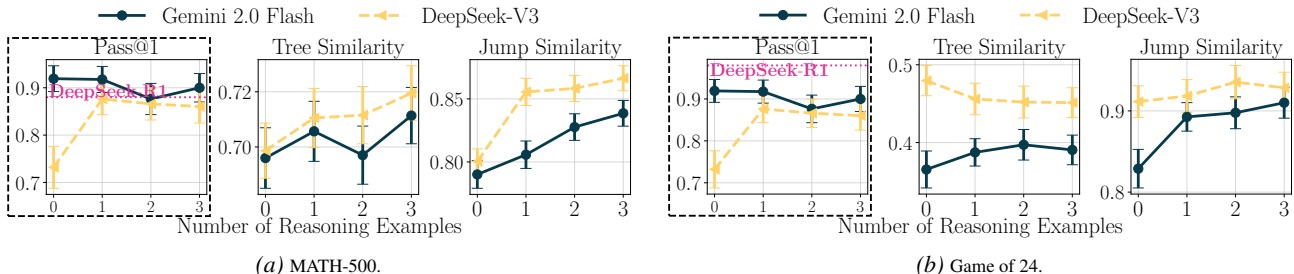

*Figure 19.* **Effect of the number of in-context reasoning examples on reasoning behaviors.** We include reasoning examples generated by DeepSeek-R1 in the prompt to guide LLMs (Gemini 2.0 Flash and DeepSeek-V3) to reason more like LRMs. The dashed boxes indicate final accuracy for different number of in-context examples, while the remaining plots show tree similarity and jump similarity to DeepSeek-R1. Neither pass@1 nor tree similarity exhibits a consistent correlation with the number of examples. In contrast, jump similarity increases nearly monotonically, suggesting that fine-grained reasoning actions (e.g., verification, calculation, backtracking) are more influenced by reasoning examples, whereas high-level problem decomposition shows no consistent change.

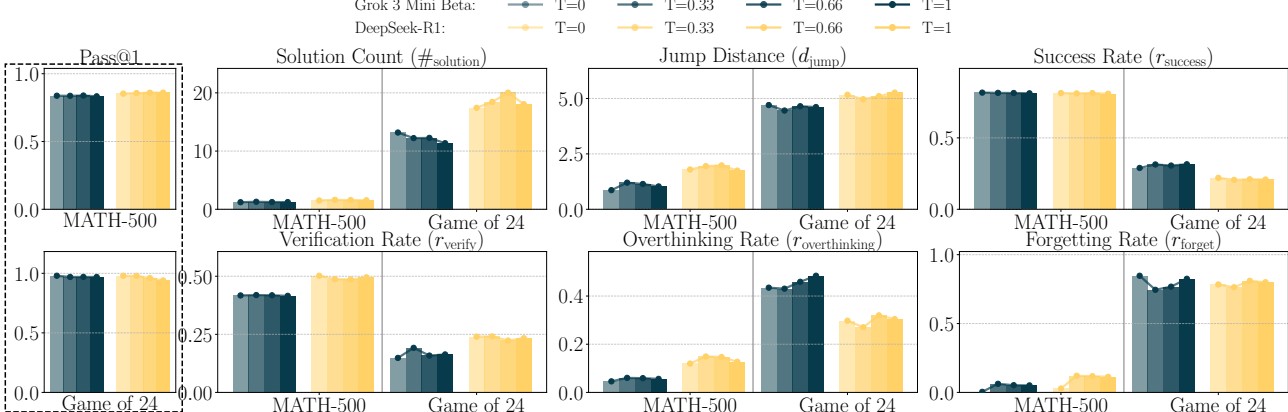

*Figure 20.* **Impact of decoding temperature on reasoning behaviors across two tasks (MATH-500 and Game of 24) using Grok 3 Mini Beta and DeepSeek-R1.** Each subplot reports one of seven metrics: pass@1, solution count, jump distance, success rate, verification rate, overthinking rate, and forgetting rate. We vary the temperature across $\{0.0, 0.33, 0.66, 1.0\}$ for each model. There is no consistent effect of temperature across models, datasets on reasoning behaviors.

