# OpenReview forum: "ReJump: A Tree-Jump Representation for Analyzing and Improving LLM Reasoning"
_ICML.cc/2026/Conference — ICML 2026 regular_

### Official Review · Reviewer_HbVk · 2026-03-11

**Soundness:** 2
**Presentation:** 2
**Significance:** 2
**Originality:** 2
**Overall Recommendation:** 3
**Confidence:** 2

**Summary:**

ReJump represents reasoning traces with two linked parts: a tree over partial solutions and a jump process over actions such as calculation, verification, and backtracking. It then uses an external Gemini-based extractor to turn free-form traces into metrics. Those metrics are used for downstream analysis and selection.

**Compliance With Llm Reviewing Policy:**

Affirmed.

**Final Justification:**

I have decided to retain my original score.

**Key Questions For Authors:**

I have raised the concerns outlined in the Weaknesses section. Addressing those points would help elevate what is already a strong paper to an even more compelling contribution. I am happy to revisit my score based on the authors' responses and the perspectives shared by fellow reviewers during the discussion phase.

**Limitations:**

Yes.

**Strengths And Weaknesses:**

**Strengths:**

1. The split between structure and traversal is a sensible design choice. It is easier to interpret than treating the whole trace as plain text. That representation decision is one of the stronger aspects of the paper.

2. The paper includes model differences, task differences, distillation, and RL. This broad view is useful. It also reflects substantial analytical effort.

3. The metric definitions are reasonably careful. The redundancy analysis is a useful extra check. I appreciated that the paper did not simply introduce many metrics without examining overlap.

4. There is at least some attempt to validate the extractor. The paper uses both automatic and human checks. It also compares against a simpler baseline.

**Weaknesses:**

1. The whole framework depends on a strong external extractor. This would be easier to accept if the validation were much stronger. At present, that dependency remains the central concern.

2. The action labels are too coarse. Many real reasoning behaviors are forced into `calc`, `verify`, or `backtrack`. This limits how much confidence I place in the finer behavioral conclusions.

3. The downstream gains are limited. Best-of-N is tested only in small search-style settings. The paper also does not compare the cost against simpler alternatives.

4. The main experiments are narrow. MATH-500 and Game of 24 are not enough for broad claims. I would want to see a more diverse task set.

5. The paper does not clearly separate itself from nearby graph-based work. It argues that trees are better. However, the supporting evidence remains limited.

6. Some claims rely on rather thin evidence. The RL discussion uses only a few model-task pairs. In addition, some figures are difficult to read.

---

> ### Author Rebuttal · Authors · 2026-03-31
>
> We appreciate the reviewer for acknowledging (1) the sensible representation design, (2) the broad experimental coverage, (3) the carefully designed metrics, and (4) the extractor validation effort. We summarize your concerns and address all of them below.
>
> ---
>
> > C1: The whole framework depends on a strong external extractor. This would be easier to accept if the validation were much stronger. At present, that dependency remains the central concern.
>
> `[NEW EXP]` We agree that extractor reliability is critical. On the 82 Game of 24 traces with ground-truth trees (Section 4.1), we additionally compute the Pearson correlation between GT-derived and extractor-derived metric values across all 82 traces for each of the 6 ReJump metrics. Concretely, for each metric m, we obtain two 82-dimensional vectors — m(GT) and m(extracted) — and compute their Pearson ρ. A high ρ indicates that the relative ordering of traces is preserved, i.e., traces that score high/low on a metric under GT trees also score high/low under extracted trees:
>
> | Metric | #solution | d_jump | r_success | r_overthink | r_verify | r_forget |
> |---|---|---|---|---|---|---|
> | **ρ (GT vs. extracted)** | 0.98 | 0.98 | 0.95 | 0.91 | 0.97 | 0.98 |
>
> All 6 metrics achieve ρ ≥ 0.91, confirming that cross-trace rankings are faithfully preserved despite graph-level pass@1 being 80–81%. This is because most extraction errors are minor (e.g., merging adjacent nodes) and do not meaningfully distort downstream metrics. Furthermore, pass@k with consensus selection already improves extraction accuracy beyond 90% (Table 2), and the extractor is a replaceable module — it can benefit from (1) more capable frontier models as they continue to improve rapidly, and (2) a fine-tuned model specifically trained for tree extraction.
>
> > C2: The main experiments are narrow. MATH-500 and Game of 24 are not enough for broad claims. I would want to see a more diverse task set.
>
> `[NEW EXP]` Our experimental coverage is broader than the main text alone may suggest. Sudoku and ZebraLogic were already included in Appendix D.1 (BoN selection); we omitted them from the main cross-model analysis because their profiles are exploration-dominant, similar to Game of 24. We now add AIME’26 (competition-level math) as a new, harder task to further strengthen diversity.
> See results in [figure](https://hackmd.io/_uploads/HJmK7fFsbx.png).
>
>
>
> > C3: The downstream gains are limited. Best-of-N is tested only in small search-style settings. The paper also does not compare the cost against simpler alternatives.
>
> ReJump's primary contribution is a diagnostic framework for understanding reasoning processes, not an inference-time optimization method. The BoN experiment validates that ReJump metrics carry actionable, process-level signal — selecting traces based on reasoning structure rather than final-answer consensus. This is fundamentally different from answer-agreement methods like majority vote, which provide no insight into *why* reasoning succeeds or fails. The two approaches are complementary, not competing: majority vote selects answers, while ReJump diagnoses reasoning quality.
>
> Regarding cost, the extraction step does require additional processing, but does not necessarily depend on proprietary models. With the rapid advancement of open-source models (e.g., Kimi K2, Qwen3), strong alternatives are freely available and continue to improve. To further increase efficiency, one could fine-tune a small, specialized model for this purpose. However, since our paper's goal is to propose a representational framework for analyzing reasoning — not to train a model — we consider this an interesting future direction rather than a current scope item.
>
> > C4: The paper does not clearly separate itself from nearby graph-based work. It argues that trees are better. However, the supporting evidence remains limited.
>
> See our response to `R-vV4p C2`.
>
> > C5: The three action categories are too coarse.
>
> See our response to `R-vV4p C1`.
>
> > C6: Some claims rely on rather thin evidence.
>
> The paper's broad claims are supported by comprehensive experiments in Sections 5.1–5.3 (5 models × 2 tasks × multiple temperatures, LLM-vs-LRM, distillation across 2 scales × 3 types). For RL (Section 5.4), the model sizes (1.7B, 8B) are well-matched to the task difficulty and already produce rich reasoning behavior; larger-scale RL training is constrained by GPU resources but the structural insights generalize. Sections 6.1–6.2 are application demos making methodological claims ("ReJump *can* be used for X"), where controlled demonstrations suffice; extended results appear in Appendix D.1–D.2.
>
> > C7: Some figures are difficult to read.
>
> We will enlarge figures and add annotations in the camera-ready version, which allows an additional page.
>
> ---
>
> **Final Note:** We appreciate your **openness to revisiting your score**, and hope our responses have addressed your concerns.

---

> > ### Author Rebuttal · Reviewer_HbVk · 2026-04-03
> >
> > Thank you for your rebuttal. After considering your responses as a whole, I have decided to maintain my original score.

---

> > > ### Author Response · Authors · 2026-04-03
> > >
> > > Thank you for reading our rebuttal. To recap, we provided:
> > >
> > > 1. **Extractor validation**: Pearson $\rho \geq 0.91$ across all metrics, Spearman $\rho$ analysis, and redundancy checks
> > > 2. **Perturbation analysis**: noise-injection on segmentation and labeling errors
> > > 3. **Additional datasets**: AIME 2026, Sudoku, ZebraLogic
> > > 4. **BoN baselines**: $d_{\text{jump}}$ vs. heuristic selection
> > > 5. **Action taxonomy completeness**: structural completeness argument
> > > 6. **Two-layer vs. flat-graph comparison**: detailed distinction
> > > 7. **Metric design clarifications**: mean vs. median rankings, redundancy analysis
> > >
> > > We notice that you selected "(b) Partially resolved — I have follow-up questions," but we did not find any follow-up questions in your response. Could you clarify which specific concerns remain unresolved? We are happy to provide additional details or experiments during the discussion period.

---

### Official Review · Reviewer_vV4p · 2026-03-13

**Soundness:** 3
**Presentation:** 3
**Significance:** 2
**Originality:** 2
**Overall Recommendation:** 4
**Confidence:** 4

**Summary:**

The paper proposes a framework named ReJump to analyse the reasoning trace of LLMs and LRMs. ReJump models each partial solution as a node of a tree. The model jumps between nodes to complete the reasoning process. The authors build an LLM agent to extract ReJump of each reasoning trace. Based on the ReJump framework, the authors propose a set of metrics to describe the characteristics of a reasoning trace. These metrics are used to analyze the reasoning structures across different models and tasks, as well as enhance model performance.

**Compliance With Llm Reviewing Policy:**

Affirmed.

**Final Justification:**

The authors propose a novel diagnostic framework to analyze the reasoning paths of LLMs and LRMs. The proposed framework is well-defined and distinguishes itself from previous work by introducing a two-layer decomposition of the reasoning paths. The authors also conduct comprehensive experiments and provide useful insights. Although requiring multiple calls of close source models, ReJump does not need to be applied to each sample, therefore keeping the cost within an acceptable budget. Generally speaking, the authors' rebuttal addresses my concerns.

**Key Questions For Authors:**

1. What is the difference between ReJump and previous graph-based frameworks?
2. How is the completeness of the action-format classification ensured?
3. How is the granularity of the partial solution within a node defined and controlled, especially for math tasks?

**Limitations:**

Yes

**Strengths And Weaknesses:**

**Strengths**
1. The experiments of the paper are comprehensive and provide solid support for the authors' findings.
2. The presentation of the paper is clear. The authors give strict definitions of the ReJump framework as well as the evaluation metrics. The important conclusions are clear and easy for readers to follow.
3. The proposed method cannot only analyse the reasoning structure, but also improve model performance by selecting prompts or responses. This provides insights for future work.

**Weaknesses**
1. The difference between ReJump and other graph-based representations requires further clarification. ReJump models the reasoning trace as a tree structure along with jumping actions, which is similar to a graph structure. I recommend making a clearer explanation of the difference between ReJump and existing graph-based analysis frameworks.
2. The classification of the action formats seems heuristic. There lacks theorems or empirical results to confirm that classifying action into three formats is complete or general enough to cover most reasoning actions.
3. The cost of the method is high. Constructing ReJump of each reasoning trace requires multiple calls to close source models, which increases the cost and makes the method difficult to use in practice.

---

> ### Author Rebuttal · Authors · 2026-03-31
>
> We thank the reviewer for the careful review, and for acknowledging (1) the comprehensive experiments, (2) the clear presentation and strict metric definitions, and (3) the practical value of prompt/response selection. We summarize your concerns and address all of them below.
>
> ---
>
> > C1: How is the completeness of the action-format classification ensured?
>
> The three categories are an exhaustive partition of tree traversal transitions, defined over structure rather than content. At any step, the model either visits a **new node** (= calculation) or an **existing node** (= verification if checking it, backtracking if branching from it). Behaviors like knowledge recall, brainstorming, and heuristic guessing all produce new partial solutions and are therefore classified as calculation — regardless of the cognitive strategy involved. The taxonomy is complete because there is no fourth structural case.
>
> Regarding whether expanding the taxonomy would change the main conclusions: the categories the reviewer suggests (knowledge retrieval, ideation, heuristic guessing) describe **what happens within a node** (the content of a reasoning step), not how the model moves between steps. These are orthogonal to our traversal-level taxonomy and would not change it. That said, finer within-node categorization is an interesting future direction that could complement ReJump's current structural analysis.
>
> > C2: What is the difference between ReJump and previous graph-based frameworks?
>
> We will incorporate the following discussion into the introduction and the "Approaches for Reasoning Analysis" paragraph in related work to clearly differentiate ReJump from existing graph-based frameworks.
>
> - **Graph-based frameworks (one layer):** Each reasoning step is a separate node in a flat graph with sequential edges. Structure and behavior are entangled in a single layer — e.g., a backward edge from step 5 to step 2 could be backtracking, verification, or forgetting, but the graph cannot distinguish them. As a result, metrics like *how far the model backtracks*, *whether it re-derives an already-solved sub-problem*, or *whether it over-explores non-solution branches* are difficult to define.
>
> - **ReJump (two layers):** The *tree layer* captures the structural hierarchy of partial solutions (which steps build on which), and the *jump layer* records the behavioral traversal (how the model navigates the tree — calculation, verification, or backtracking). This 2D decomposition makes metrics naturally definable: d_jump uses tree depth to measure backtrack distance; r_forget uses node identity to detect re-derivation; r_overthink uses the solution path to identify wasted exploration. Two traces arriving at the same answer (same tree) but via different exploration strategies (different jumps) are directly comparable — something a flat graph cannot express.
>
> > C3: How is the granularity of the partial solution within a node defined and controlled, especially for math tasks?
>
> The extraction prompt (Appendix B.1) defines a node as a step that can be stated as an independent mathematical sub-problem with a clear **Problem** (input) and **Result** (output) — analogous to a lemma in a proof. Steps that cannot be independently formulated (e.g., algebraic manipulations, substitutions) are merged into their parent node. For example, in a MATH problem requiring solving a quadratic equation within a larger proof, "solve x² − 5x + 6 = 0 → x = 2 or x = 3" forms a single node, while the intermediate factoring steps are merged into it. Our human evaluation (pass@1 = 80–81%) confirms that this criterion is reproducible: human annotators and the LLM extractor agree on node boundaries at a high rate, indicating the granularity standard is well-defined enough to be consistently applied.
>
> > C4: The cost of the method is high. Constructing ReJump of each reasoning trace requires multiple calls to close source models, which increases the cost and makes the method difficult to use in practice.
>
> ReJump is a *diagnostic* tool — a practitioner only needs to run it on a small set of representative traces to identify systematic reasoning patterns, not on every sample exhaustively. The extractor is a replaceable module not tied to any specific model; open-source alternatives (e.g., Qwen3, Kimi K2) nowadays also work well and can serve as the extractor at lower cost. To reduce complexity, a smaller model fine-tuned specifically for tree extraction is a natural next step — the extraction task (segmenting steps and labeling transitions) is well-defined and structurally constrained, making it well-suited for supervised fine-tuning. Our goal in this work is to propose the representation and analysis framework. Training a dedicated extractor is a very interesting future direction.
>
> ---
>
> **Final Note:** We hope we have addressed your concerns. If our responses have resolved your concerns, we kindly request you to consider increasing your score and championing our paper.

---

> > ### Author Rebuttal · Reviewer_vV4p · 2026-04-03
> >
> > I thank the authors for carefully explaining the completeness of the action-format classification. Overall, the authors' rebuttal has addressed my concerns, and I would like to raise my recommendation to 4.

---

> > > ### Author Response · Authors · 2026-04-03
> > >
> > > Thank you for the careful re-evaluation and for raising your score. We appreciate the constructive feedback throughout the review process, which has helped strengthen the paper. We will incorporate the remaining suggestions in the camera-ready version. Thanks again for your time and effort.

---

### Official Review · Reviewer_eank · 2026-03-14

**Soundness:** 4
**Presentation:** 4
**Significance:** 3
**Originality:** 3
**Overall Recommendation:** 6
**Confidence:** 5

**Summary:**

The paper proposes **ReJump**, a structured representation for reasoning traces that models an LLM’s chain-of-thought as traversal over a tree of intermediate states, including both local transitions and long-range jumps intended to capture behaviors such as backtracking and verification. Building on this representation, the authors define a set of reasoning metrics and introduce a ReJump-Extractor that maps raw traces into the proposed format. Empirically, the method is evaluated on Game of 24 and MATH-500, where the extractor appears to achieve reasonably strong agreement with human annotation, and the resulting analysis is used to compare reasoning patterns across contemporary reasoning models beyond final-answer accuracy alone. The paper further studies the effects of prompting, distillation, and reinforcement learning on these patterns, and reports that ReJump-based test-time selection can improve performance in Best-of-N and prompt-selection settings on Game of 24. Overall, the work is positioned as both an interpretability framework for analyzing reasoning structure and a practical tool for improving inference-time decision making.

**Compliance With Llm Reviewing Policy:**

Affirmed.

**Final Justification:**

The authors have fairly addressed my concerns with additional experiments.

**Key Questions For Authors:**

1.  **How sensitive are the main conclusions to the choice of step categories?**
    Would the comparative findings still hold if the taxonomy were expanded to include behaviors such as knowledge retrieval, ideation, heuristic guessing, or hypothesis generation?

2.  **What kinds of reasoning traces are not well captured by the current representation?**
    A discussion of known failure cases or missing categories would help clarify the current scope of the method.

3.  **Would adding richer categories improve downstream usefulness, or mainly increase annotation complexity?**
    This is important for understanding whether the current taxonomy is a reasonable simplification or a real bottleneck.

4.  **How robust are the reported metrics to noise in segmentation or labeling?**
    For example, if a fraction of intermediate steps are mislabeled or boundaries are shifted slightly, do the high-level conclusions remain stable?

5.  **Can the authors provide an ablation or perturbation analysis on extractor errors?**
    It would strengthen the paper to simulate imperfect extraction and measure how much the downstream metrics and model rankings change.

6.  **Which part of the pipeline is most error-sensitive: segmentation, label assignment, or jump detection?**
    Understanding where errors matter most would help clarify the practical reliability of the framework.

**Strengths And Weaknesses:**

**Strengths**

1. One of the nicest aspects of the work is that it focuses on the process of reasoning rather than only outcome-level accuracy. The proposed representation makes it easier to see how a model explores, revises, and verifies intermediate thoughts, which is genuinely helpful for understanding CoT behavior.
2. I like that the paper goes beyond interpretability for its own sake and shows that the proposed analysis can actually help at test time, for example by choosing better samples or prompts. That makes the contribution feel more substantial and actionable.
3. Modeling reasoning traces as a tree with local transitions and occasional jumps is a fairly natural abstraction. It is simple enough to be understandable, but still expressive enough to capture interesting differences in reasoning style.
4.  A strong point of the paper is that it shows how models with similar benchmark performance can still behave quite differently internally. That is a valuable insight, and it supports the broader motivation for process-level evaluation.



**Weaknesses**

1.    The proposed categories capture some important reasoning behaviors, but they do not seem to cover the full range of what appears in real reasoning traces. For instance, traces often involve things like recalling relevant knowledge, brainstorming possibilities, or making plausible guesses under uncertainty, and these behaviors do not seem to fit neatly into the current taxonomy.
   2.   A practical concern is that the framework relies on an external model to identify and label intermediate steps. This adds cost and complexity, and it may make the method harder to use at scale or in settings where access to strong models is limited.
 3. Since the proposed metrics depend on how well the intermediate reasoning structure is recovered, mistakes in segmentation or labeling could affect the final analysis. I would have liked to see a bit more discussion of how robust the findings are to imperfect extraction.

---

> ### Author Rebuttal · Authors · 2026-03-31
>
> We thank the reviewer for the thoughtful review, and for recognizing (1) the focus on reasoning process rather than outcome-level accuracy, (2) the actionable test-time applications, (3) the natural tree-jump abstraction, and (4) the insight that models with similar accuracy can behave differently internally. We summarize your concerns and address all of them below.
>
> ---
>
> > C1: How robust are the metrics to noise in segmentation or labeling? Which part of the pipeline is most error-sensitive?
>
> `[NEW EXP]` We answer this directly via a perturbation analysis. Starting from the 82 ground-truth Game of 24 trees, we inject three types of controlled noise at 5% and 10% rates and measure per-metric relative error. The three perturbation types simulate realistic extraction errors: (1) **node merge**: randomly merge two adjacent sibling nodes into one, simulating under-segmentation where the extractor fails to separate two distinct reasoning steps; (2) **node split**: randomly split one node into two children, simulating over-segmentation; (3) **action label flip**: randomly change a transition's action label (e.g., calculation → verification), simulating misclassification of the jump type.
>
> | Perturbation | #solution | d_jump | r_success | r_verify | r_overthink | r_forget |
> |---|---|---|---|---|---|---|
> | 5% node merge | 2.4 | 3.3 | 1.7 | 9.0 | 4.9 | 2.8 |
> | 10% node merge | 5.3 | 10.8 | 2.2 | 9.4 | 10.0 | 2.8 |
> | 5% label flip | 0.0 | 0.0 | 0.0 | 8.0 | 0.0 | 0.0 |
> | 10% label flip | 0.0 | 0.0 | 0.0 | 14.7 | 0.0 | 0.0 |
> | 5% node split | 0.0 | 3.5 | 0.0 | 6.3 | 0.0 | 0.0 |
> | 10% node split | 0.0 | 7.2 | 0.0 | 7.5 | 0.0 | 0.0 |
>
> *Values are mean relative error (%) = |perturbed − original| / |original| × 100, averaged over 10 random trials on 82 Game of 24 ground-truth trees.*
>
> **Label flip** affects only r_verify (up to 14.7%); all other metrics are exactly invariant since they do not depend on action labels. **Segmentation** errors have broader but modest impact — even in the worst case (10% node merge), all metrics stay below 11% relative error. Overall, segmentation is the more error-sensitive component, but the error magnitudes are small enough that downstream conclusions remain stable under realistic noise levels.
>
> > C2: The three action categories do not seem to cover the full range of reasoning behaviors (e.g., knowledge recall, brainstorming, heuristic guessing). Would expanding the taxonomy change the main conclusions, or mainly increase annotation complexity?
>
> See `R-vV4p C1`.
>
> > C3: What kinds of reasoning traces are not well captured by the current representation? A discussion of known failure cases or missing categories would help clarify the current scope of the method.
>
> There should not be reasoning traces that fall outside ReJump's representational scope. The tree+jump representation is structurally complete for sequential reasoning: every step either derives a new partial solution (new tree node, labeled calc) or revisits an existing one (verify or backtrack), and the jump layer records the full traversal order. There is no fourth case.
>
> Regarding specific patterns: "cycles" (the model circling back to the same idea) are captured as revisits to existing nodes — either verification or forgetting (r_forget). "Mutual dependencies" (two sub-problems informing each other) are represented as sibling nodes sharing a common parent — the dependency is structural, and the tree captures it directly.
>
> > C4: A practical concern is that the framework relies on an external model to identify and label intermediate steps. This adds cost and complexity, and it may make the method harder to use at scale or in settings where access to strong models is limited.
>
> See `R-vV4p C4`.
>
> ---
>
> **Final Note:** We thank you again for your support and constructive questions. If there are any remaining questions, please do not hesitate to let us know.

---

> > ### Author Rebuttal · Reviewer_eank · 2026-04-06
> >
> > Thank you for the thorough clarification and for providing the additional experiments. These address my concerns satisfactorily. I have decided to increase my score by 1.

---

> > > ### Author Response · Authors · 2026-04-06
> > >
> > > Thank you for the careful re-evaluation and for raising your score. We are grateful for your strong support and for championing our paper. We will incorporate your earlier suggestions in the camera-ready version. Thanks again for your time and constructive feedback throughout the review process.

---

### Official Review · Reviewer_srZA · 2026-03-15

**Soundness:** 3
**Presentation:** 3
**Significance:** 2
**Originality:** 3
**Overall Recommendation:** 4
**Confidence:** 2

**Summary:**

This paper introduces ReJump, a two-layer representation for analyzing LLM reasoning traces. The first layer encodes the reasoning structure as a tree of intermediate problem-solving steps, and the second captures the visitation order over that tree (the "jump layer"), with transitions labeled as calculation, verification, or backtracking. The authors define six evaluation metrics (solution count, jump distance, success rate, verification rate, overthinking rate, forgetting rate) and two comparison metrics (tree and jump similarity). They build an LLM-based extractor (ReJump-Extractor) using Gemini 2.5 Pro to parse reasoning traces into this format, validate it on Game of 24 and MATH-500, and then use it to compare reasoning behaviors across models, training methods, and tasks. Finally, they show ReJump can improve test-time performance via Best-of-N selection and prompt selection.

**Compliance With Llm Reviewing Policy:**

Affirmed.

**Key Questions For Authors:**

1. How sensitive are the cross-model comparison results to extraction errors? If ReJump-Extractor makes systematic errors on certain models (e.g., misclassifying backtracking as calculation for longer traces), could this explain some of the observed behavioral differences?
2. For the BoN experiments, have you compared selecting by d_jump against simpler heuristics like selecting by trace length or number of distinct solution attempts?
3. How would ReJump handle tasks where the reasoning structure is not naturally tree-like (e.g., reasoning with cycles or mutual dependencies)?

**Limitations:**

yes

**Strengths And Weaknesses:**

Strengths
1. Well-motivated formalism. The core idea of decomposing reasoning into a structural layer (tree) and a behavioral layer (jump) is clean and intuitive. The paper makes a convincing case that final-answer accuracy alone is insufficient for understanding reasoning models, and the tree-jump abstraction provides a natural vocabulary for discussing exploration, exploitation, verification, and backtracking. The visual examples in Figures 1 and 3 are effective at communicating the representation.
2. Breadth of analysis. The paper covers an impressive range of comparisons: across models (five LRMs), across tasks (MATH-500 vs. Game of 24), LLMs vs. LRMs, distilled vs. teacher models, and RL training dynamics. Several of the findings are informative — for instance, that models with similar accuracy can have very different reasoning profiles, that RL progressively aligns reasoning behavior with task demands, and that distillation transfers reasoning style but not necessarily success rate.
3. Practical utility demonstrated. The Best-of-N and prompt selection experiments (Sec. 6) show that ReJump metrics can be used without ground-truth labels to improve performance, which is a meaningful contribution beyond pure analysis.
4. Thorough appendix. The paper includes extensive supplementary material: full extraction prompts, example reasoning traces with corresponding ReJump visualizations, redundancy analysis, prompt sensitivity experiments, and additional datasets (Sudoku, ZebraLogic).

Weaknesses
1. Heavy reliance on a black-box extractor. The entire framework depends on ReJump-Extractor, which itself is a prompted LLM (Gemini 2.5 Pro). This creates a circularity concern: the paper analyzes LLM reasoning using another LLM's interpretation of that reasoning. The automatic evaluation (Table 1) is conducted on a small synthetic dataset of only 82 traces for Game of 24 with controlled properties, which may not reflect the difficulty of parsing real, long, messy reasoning traces. The human evaluation on MATH-500 (Table 2) shows pass@1 of only 80–81%, meaning roughly one in five extractions is incorrect. For a measurement tool, this error rate is concerning and could meaningfully affect downstream metric calculations and cross-model comparisons. The paper would benefit from a more rigorous error analysis examining how extraction failures correlate with specific metric distortions.
2. Limited task diversity. Despite the generality of the framework, the core experiments use only two tasks: MATH-500 and Game of 24. While the appendix adds Sudoku and ZebraLogic for the BoN experiments, the main analysis and most findings rest on just two benchmarks. Game of 24 is a relatively narrow combinatorial task, and MATH-500, while diverse in mathematical topics, has a particular structure. It remains unclear how well the three-category action taxonomy (calculation, verification, backtracking) transfers to other reasoning domains such as code generation, commonsense reasoning, or multi-hop QA, where the boundaries between these categories may be far less clear.
3. Metric design choices lack strong justification. The six metrics are reasonable but somewhat ad hoc. For example, jump distance averages tree distances between consecutive derived solution steps, but why this particular aggregation? Why not median, max, or a distribution-level comparison? The forgetting metric is binary per trace and then averaged, losing granularity. The overthinking metric counts all leaf visits after the first correct solution, but doesn't distinguish between useful verification of that solution and genuinely wasteful exploration. The redundancy analysis (Sec. A.2) shows that several metrics have high redundancy scores (e.g., overthinking at 0.944 on MATH-500), suggesting they may not provide as much independent signal as implied.
4. Similarity metrics are coarse. As the authors themselves acknowledge (Sec. 3.2), tree similarity ignores node semantics entirely, and jump similarity reduces the entire action sequence to a 3×3 transition probability matrix, discarding all temporal information. Two models could have identical transition matrices but completely different reasoning flows (e.g., all verification at the end vs. interleaved). This is a significant limitation for a representation that is supposed to enable fine-grained comparison.
5. The BoN and prompt selection improvements are modest and narrowly demonstrated. The BoN results (Table 5) show improvements of 4–7 percentage points on Game of 24, but only for two models and with N=3. The comparison baseline is majority voting, but other BoN selection criteria (e.g., length, self-consistency variants, or reward-model scoring) are not compared. The prompt selection experiment (Table 6) tests only four hand-designed prompts on a single task. Without comparing to other prompt selection methods or demonstrating gains on diverse tasks, the practical significance is difficult to assess.
6. Causal claims from correlational evidence. The feature importance analysis (Table 3) uses XGBoost to relate metrics to correctness, but this is correlational. Saying that Game of 24 "favors exploration" based on the importance of d_jump is a reasonable hypothesis, but the paper sometimes presents such findings as if they are causal or prescriptive (e.g., using high d_jump as a selection criterion in BoN). The BoN results on Sudoku/ZebraLogic, where low d_jump is selected instead, partially address this, but the framework for deciding which metric to optimize for a new task remains unclear without labeled data.
7. RL experiment is underpowered. The RL training analysis (Sec. 5.4) uses Qwen3-1.7B for Game of 24 and Qwen3-8B for MATH-500 — notably smaller models than those in the main comparison. Using different model sizes for different tasks makes it hard to disentangle model-scale effects from task effects. Only four checkpoints are evaluated, providing limited resolution on training dynamics.

---

> ### Author Rebuttal · Authors · 2026-03-31
>
> We thank the reviewer for the encouraging feedback. We summarize your concerns and address all of them below.
>
> ---
> > C1: Three concerns about ReJump-Extractor: (1) circularity of using an LLM to interpret another LLM; (2) automatic evaluation on only 82 synthetic traces; (3) ~20% error rate could distort metrics and cross-model comparisons.
>
> 1. **No circularity.** The extractor *parses* (segments steps, labels transitions) — it does not reason about the underlying problem. Since it serves an entirely different purpose from the target model, there is no circularity.
>
> 2. **Supported by large-scale human evaluation.** Beyond automatic evaluation, human evaluation on MATH-500 (Table 2) covers 500 prompts × 2 models = 1,000 real traces, providing a reliable accuracy estimate.
>
> 3. `[NEW EXP]` **The ~20% extraction error rate has limited impact on metrics.** We quantify how much ~20% extraction error actually affects each metric. The results are reported in [table](https://hackmd.io/_uploads/Hk977ftoWg.png).
>
>     A noise-injection perturbation analysis (5%/10% rates) further confirms that metrics remain stable even when additional noise is introduced (see `R-eank C1` for full results).
>
> > C2: Metric design concerns: (1) why mean over median/max for jump distance; (2) binary forgetting loses granularity; (3) overthinking conflates verification with waste; (4) high redundancy suggests limited independent signal; (5) similarity ignores semantics.
>
> `[NEW EXP]` **(1)** We verified that mean and median produce the same cross-model rankings (Spearman ρ = 0.95 on both benchmarks), so aggregation choice does not affect our conclusions. Will add to camera-ready.
>
> **(2)** Forgetting (re-deriving a visited leaf) is rare — most traces have zero occurrences, so per-step frequency would be dominated by zeros. Binary detection is natural at the trace level; population-level granularity is recovered by averaging across traces.
>
> **(3)** Overthinking already excludes verification. r_overthink counts only derivation steps reaching a leaf via *calc* after the first correct solution (Sec 3.1); verification visits are explicitly excluded and captured by r_verify.
>
> **(4)** Redundancy is task-dependent: on easy MATH-500, short traces cause co-variation (r_overthink redundancy = 0.944); on harder Game of 24 each metric captures more independent signal (drops to 0.840; r_verify shifts from 0.105 to 0.687). We include this analysis to transparently show where metrics overlap vs. diverge.
>
> **(5)** Acknowledged in Section 7. Our metrics target *structural* reasoning patterns (exploration style, backtracking). Semantic matching would require per-node-pair LLM inference — prohibitively expensive and beyond our scope of structural analysis.
>
> > C3: BoN gains are modest and narrow (2 models, N=3); no comparison to simpler heuristics or reward-model baselines.
>
> BoN and prompt selection are **proof-of-concept demonstrations**, not the paper's primary contribution.
>
> `[NEW EXP]` **d_jump outperforms simpler heuristics:**  We compare BoN selection using d_jump against two intuitive baselines — selecting the longest trace and selecting the trace with the most solution attempts — as well as majority vote (N=3, Game of 24). See results in [table](https://hackmd.io/_uploads/HkU8Xzti-e.png).
>
>
> **On reward-model baselines:** Reward models require task-specific labeled data and model training. ReJump selection needs only a lightweight task-level prior — no training or per-sample annotations.
>
> > C4: Only two main benchmarks.
>
> See `R-HbVk C2`.
>
> > C5: Does the action taxonomy transfer to other reasoning domains?
>
> See `R-eank C2`.
>
> > C6: Non-tree-like reasoning structures (cycles, mutual dependencies)?
>
> See `R-eank C3`.
>
> > C7: Feature importance is correlational; unclear how to choose metrics for new tasks.
>
> The BoN experiment shows these correlational features carry actionable signal — e.g., selecting traces with higher d_jump improves accuracy on exploration-heavy tasks (Table 5). The optimal metric varies by task (high d_jump for Game of 24 vs. low for Sudoku/ZebraLogic), itself a ReJump finding. For new tasks, a small labeled sample suffices via feature importance analysis (Table 3).
>
> > C8: RL experiment is underpowered (small models, few checkpoints).
>
> Section 5.4 demonstrates ReJump as an analytical tool for tracking reasoning behavior during RL — not a comprehensive RL study. The key finding (RL progressively aligns behavior with task-specific demands) is consistent across both model–task pairs. Different model sizes reflect GPU constraints, and the conclusion concerns behavioral adaptation, not model scale.
>
> ---
>
> **Final Note:** We hope we have addressed your concerns. If there are any remaining questions, please do not hesitate to let us know. If our responses have resolved your concerns, we kindly request you to consider increasing your score and championing our paper.

---

> > ### Author Rebuttal · Reviewer_srZA · 2026-04-04
> >
> > Thank you for the detailed and thorough rebuttal. After careful consideration, I have decided to maintain my original scores.

---

> > > ### Author Response · Authors · 2026-04-04
> > >
> > > Thank you for confirming that our responses have fully addressed your concerns, and for your support in accepting our paper. We appreciate the constructive feedback throughout the review process and will incorporate your suggestions in the camera-ready version.

---

### Decision · Program_Chairs · 2026-04-30

**Decision:**

Accept (regular)

**Comment:**

ReJump represents reasoning traces as a visitation order over nodes in a tree of intermediate problem-solving steps, enabling diverse metrics that quantify exploration, exploitation, overthinking, forgetting, and verification. The authors evaluate state-of-the-art LRMs on two tasks and find that models with similar accuracy can exhibit distinct reasoning behaviors, while different tasks favor different reasoning styles (e.g., varying balance between exploration and exploitation). The authors also show that ReJump can improve reasoning quality via ReJump-guided Best-of-N selection and prompt selection.

Reviewers appreciated the authors' core contribution; a clear and well-motivated process-level framework for analyzing reasoning traces and found the experiments comprehensive and informative, especially the result that models with similar accuracy can still exhibit very different reasoning styles.

Overall the meta-reviewer recommends this paper for acceptance, but also suggests that the authors consider seriously the outstanding reviewer comments -- in particular, if the camera-ready version included more comprehensive supplementary experiments, for example on code-related or more general tasks, this would likely improve the paper's impact.